# Transformation of tenofovir into stable ProTide nanocrystals with long-acting pharmacokinetic profiles

Denise A. Cobb[1], Nathan Smith[1], Suyash Deodhar [1], Aditya N. Bade[1], Nagsen Gautam[2], Bhagya Laxmi Dyavar Shetty[1], JoEllyn McMillan[1], Yazen Alnouti[2], Samuel M. Cohen [3], Howard E. Gendelman [1,2] & Benson Edagwa [1✉]

Treatment and prevention of human immunodeficiency virus type one (HIV-1) infection was transformed through widespread use of antiretroviral therapy (ART). However, ART has limitations in requiring life-long daily adherence. Such limitations have led to the creation of long-acting (LA) ART. While nucleoside reverse transcriptase inhibitors (NRTI) remain the ART backbone, to the best of our knowledge, none have been converted into LA agents. To these ends, we transformed tenofovir (TFV) into LA surfactant stabilized aqueous prodrug nanocrystals (referred to as NM1TFV and NM2TFV), enhancing intracellular drug uptake and retention. A single intramuscular injection of NM1TFV, NM2TFV, or a nanoformulated tenofovir alafenamide (NTAF) at 75 mg/kg TFV equivalents to Sprague Dawley rats sustains active TFV-diphosphate (TFV-DP) levels ≥ four times the 90% effective dose for two months. NM1TFV, NM2TFV and NTAF elicit TFV-DP levels of 11,276, 1,651, and 397 fmol/g in rectal tissue, respectively. These results are a significant step towards a LA TFV ProTide.

[1] Department of Pharmacology and Experimental Neuroscience, University of Nebraska Medical Center, Omaha, NE, USA. [2] Department of Pharmaceutical Sciences, University of Nebraska Medical Center, Omaha, NE, USA. [3] Department of Pathology and Microbiology, University of Nebraska Medical Center, Omaha, NE, USA. ✉email: benson.edagwa@unmc.edu

Despite advances made by the broad use of antiretroviral therapy (ART) for the treatment and prevention of human immunodeficiency virus type one (HIV-1) infection adherence to daily regimens remains a major obstacle in achieving sustained viral suppression and in preventing viral transmission[1]. It is generally perceived that extended-release long-acting (LA) ART designed to ease patient dosing and synchronize CD4+ T cell and HIV-1 load measurements could positively impact treatment outcomes. Notably, LA formulations are widely used as contraceptives and antipsychotics and have already shown improvement in regimen adherence[2,3]. This idea has recently been forged for HIV-1 therapy as rilpivirine and cabotegravir (RPV and CAB) LA are now approved for human use[4,5] in Canada, Europe, and the US[6] as injectable solid drug nanosuspensions. The injectables are administered monthly or every other month[7,8]. Such LA antiretroviral drug (ARV) delivery strategies could serve to benefit the most vulnerable populations such as intravenous drug users, children, and pregnant women. However, limitations of CAB and RPV LA include the requirement for large injection volumes, short dosing intervals, none of them are active against major HIV co-infections such as hepatitis B and are also characterized by limited drug access to cellular and tissue infectious reservoirs[9–11]. Attempts to overcome these have led to the transformation of native ARVs into nanoformulated prodrugs designated as slow effective release antiretroviral therapy (LASER ART). These are homogeneous dispersions of solid prodrug nanocrystals stabilized by aqueous surfactant solutions[12–15]. The LASER ART prodrug nanocrystals provide slow drug dissolution and rapid transport across biological membranes. This can achieve sustained plasma and tissue drug concentrations above the 90% effective concentration (EC90) for periods of up to one year[16].

The target product profile of compounds suitable for parenteral LA formulations includes high potency, lower injection volumes, extended dosage intervals, low aqueous solubility, drug stability, slow dissolution and release, and extended elimination times. All can be achieved, in part, through the creation of prodrugs. For example, the ProTide prodrug technology has enabled safe and efficient intracellular delivery of phosphorylated nucleoside and nucleotide analogs in virus target cells[17]. The application of ProTide technology to tenofovir (TFV) resulted in the discovery of TFV alafenamide (TAF), a phosphoramidate prodrug. Compared to tenofovir disoproxil fumarate (TDF), TAF is more potent and its safety profile makes it a better candidate for transformation into an LA formulation[18,19]. However, TAF is susceptible to hydrolytic degradation, by non-enzymatic hydrolysis in an aqueous environment[20,21]. Attempts to modify formulation excipients or pH to affect TAF stability are yet to demonstrate improved pharmacokinetic (PK) outcomes[22–25]. We now describe an improved ProTide strategy that masks the phosphate groups in TFV with cleavable hydrophobic amino acid ester lipids with synergistic effects on nucleoside analogs[26,27]. These modifications affect drug efficacy, prodrug aqueous stability, dissolution, half-life, and safety[26–28].

In this work, we describe the preclinical development of two lipophilic aqueous stable TFV ProTide nanoformulations referred to as NM1TFV and NM2TFV. The formulations exhibit improved intracellular drug delivery, retention, and protection against HIV-1 infection in vitro when compared against a nanoformulated TAF control (NTAF). Notably, a single intramuscular injection of NM1TFV or NM2TFV to Sprague Dawley (SD) rats at 75 mg/kg TFV equivalents sustains TFV-DP levels above the EC90 in peripheral blood mononuclear cells (PBMC) for two months with no recorded adverse events. The lead NM1TFV sustains tissue TFV-DP levels four times higher than the known drug EC90 of 16−48 fmol/10^6 cells linked to HIV-1

prevention[29]. Our studies provide proof of concept that conversion of TFV into stable ProTide formulations can provide extended drug dosing intervals.

## Results

**Synthesis and characterization of TFV ProTides.** We created lipophilic ProTides of TFV bearing phenylalanine and alanine amino acid esters (coined as M1TFV and M2TFV) by replacing optimal short chain alkyl ester groups utilized by conventional ProTide strategies[28,30,31]. Bulky residues were used to create the ProTides (Fig. 1). The docosanol masking ester motif was selected based on inherent lipophilicity and synergy with nucleoside analogs[26,27]. M1TFV and M2TFV were synthesized by coupling a phenylalanyl or alanyl docosyl ester (Fig. 1a, b) to monophenyl TFV in the presence of Et3N to yield the desired compounds with chemical yields of 50–65% and purities of 98.4% for M1TFV, and 94.9% for M2TFV (Supplementary Figs. 1−3). Since the conjugation step is moisture sensitive, further improvements to the chemical yields could be achieved by either optimizing the coupling reagents or reaction vessels. Infusion of the synthesized compounds into an Autoflex maX MALDI-TOF/TOF mass spectrometer confirmed molecular mass ions of 477.022 for TAF, 819.42 for M1TFV, and 743.361 for M2TFV (Supplementary Figs. 4−6). Conversion of TFV into ProTides altered the physicochemical properties of TFV. TAF, M1TFV and M2TFV exhibited aqueous solubilities of 5.52, 0.002, and 0.007 mg/mL, respectively. In contrast, the solubility of the prodrugs in 1-octanol were 4.85, 29.2, and 20.9 mg/mL for TAF, M1TFV, and M2TFV, respectively. This confirmed increases in prodrug lipophilicity (Fig. 1c). To determine the effect of prodrug modifications and nanoparticle dissociation on compound chemical stability, 1 μM NM1TFV or NTAF formulations were incubated for 24 h in buffers of increasing pH. These included pH of 2.0 (0.1% formic acid), 6.0 (7.5 mM ammonium acetate), 7.0 (PBS), 8 (7.5 mM ammonium bicarbonate), and 10.3 (0.1% ammonium hydroxide). No statistical differences in prodrug stability upon nanoparticle dissociation and prodrug dissolution were observed between NM1TFV and NTAF at pH 2.0 from the 0 to 2 h timepoints, at pH 6.0 from the 0 to 12 h timepoints, or at pH 7.0 at any timepoint (Supplementary Fig. 7 and Supplementary Table 1). However, dissociated NM1TFV demonstrated significantly greater prodrug stability at pH 2.0 at the 6 and 12 h timepoints, pH 6.0 at the 24 h timepoint, and at pH 8.0 and 10.3 at all timepoints (Supplementary Fig. 7 and Supplementary Table 1), suggesting that unlike TAF that is hydrolytically unstable at basic pH, M1TFV confers enhanced prodrug stability over a broader pH range. It is also worth noting that stability tests were performed on dissociated formulation particles upon dilution and that M1TFV is stable within NM1TFV solid drug nanoparticles. The two ProTides were further characterized by [1]H and [31]P NMR. The triplet at 0.87 ppm and multiplets at 1.21–1.35 ppm on the spectrum of M1TFV correspond to the terminal methyl and methylene protons of the docosyl ester, respectively. Chemical shifts at 7.04–7.38 ppm represent the aryl and phenylalanine masking promoieties (Fig. 1e). For M2TFV, the triplet at 0.87 ppm and multiplets at 1.20–1.38 ppm correspond to the terminal methyl and methylene protons of the docosyl ester, respectively. The triplets at 7.10, 7.21, 7.31, and doublets at 6.99 and 7.14 ppm represent the aryl and phenol groups (Fig. 1e). The two peaks in the phosphorous NMR spectra indicated a 1:1 mixture of the R and S diastereomers at the phosphorous chiral center[32] (Supplementary Fig. 8). Even though phosphorous chirality of ProTides has been shown to affect potency, our studies used the mixture of isomers since the chirality would be lost following

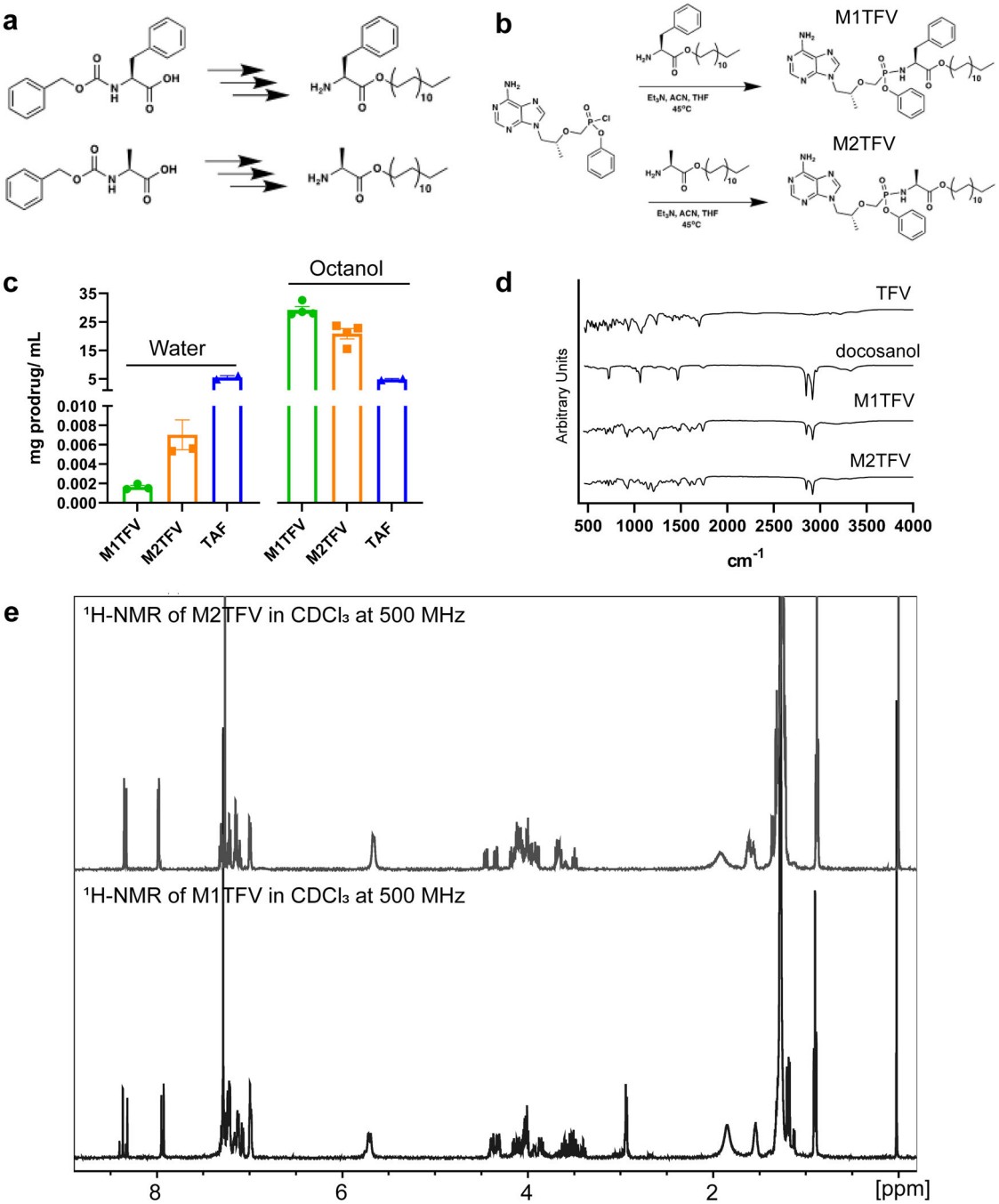

**Fig. 1 Characterization of TFV ProTides. a** Synthesis of phenyl alanine and alanine derivatizing promoieties. **b** Conjugation of the amino acid esters to activated monophenyl TFV to form M1TFV and M2TFV prodrugs. **c** Aqueous and octanol solubility of M1TFV and M2TFV demonstrate the decreased aqueous solubility of M1TFV (green) and M2TFV (orange), and increased octanol solubility compared to TAF (blue). Values reported are the mean ± SEM of three replicates. **d** FT-IR Absorption bands at 2,844 and 2,912 $cm^{-1}$ illustrate the asymmetric and symmetric C-H stretches from the long chain fatty acid. **e** An overlay of the proton NMR spectra confirmed the synthesis of M1TFV (below) and M2TFV (above). Specifically, the triplet peak at 0.87 ppm and multiplets at 1.21–1.35 ppm correspond to the terminal methyl and methylene protons of the docosyl ester on M1TFV, respectively. Chemical shifts at 7.04−7.38 ppm represent the aryl and phenylalanine masking promoieties. For M2TFV, the triplet at 0.87 ppm and multiplets at 1.20–1.38 ppm correspond to the terminal methyl and methylene protons of the docosyl ester, respectively. Chemical shifts at 6.99, 7.10, 7.14, 7.21, and 7.31 ppm represent the aryl masking promoiety. **a−e** Experiments were repeated independently five times with equivalent results.

intracellular cleavage of the masking groups. Further chemical characterization of the ProTides by FTIR showed absorption bands at 2,844 and 2,912 $cm^{-1}$ corresponding to asymmetric and symmetric C−H stretches from the long-chain fatty alcohol (Fig. 1d).

**Characterization of nanoformulated TFV ProTides**. Nanocrystal formulation strategy has so far resulted in the successful development of several monthly IM injectable products that include CAB and RPV LA. Nanocrystals provide enhanced drug dissolution, high loading while obviating organic solvent usage

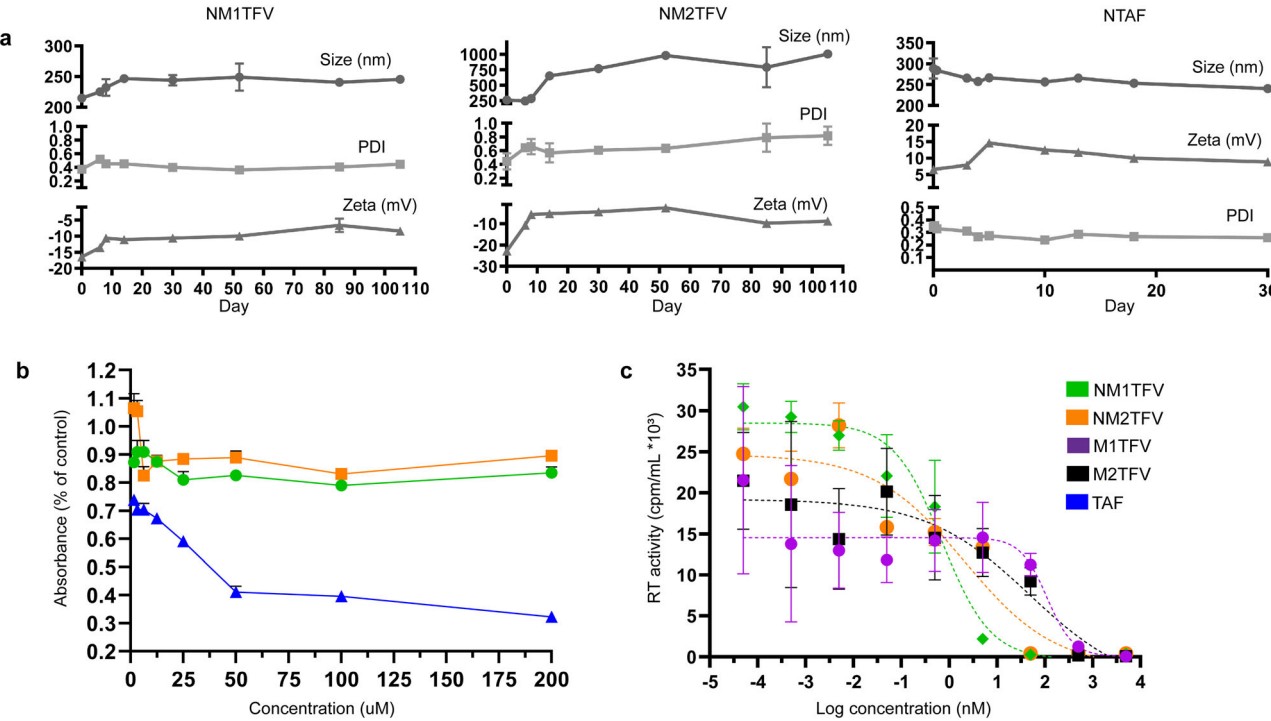

**Fig. 2 Nanoformulation of TFV ProTides.** Nanoformulations were synthesized by high-pressure homogenization using poloxamer 407 (P407) as the excipient for NM1TFV/NM2TFV, or P407 and PEG 3350 as excipients for NTAF. **a** Formulation stability at 25 °C (up to 110 days) was measured by particle hydrodynamic diameter (size), polydispersity index (PDI), and zeta potential as determined by dynamic light scattering (DLS). Data were independently reproduced three times. **b** Cell viability was assessed in MDM by MTT assay 8 h after NM1TFV, NM2TFV, or TAF treatment over a range of concentrations (1–200 μM). Results were normalized to untreated control cells. All results are shown as the mean ± SEM of four replicates. **c** $IC_{50}$ of M1TFV, M2TFV, and their nanoformulations in MDM was investigated at a range of concentrations (0.00005–5,000 nM) and determined by HIV-1 reverse transcriptase (RT) activity after viral challenge with HIV-1$_{ADA}$ at an MOI of 0.1. Data were normalized and expressed as a percentage of HIV-1 infected vehicle control ± SEM; $n = 4$ biological replicates. Experiments **b**, **c** were repeated independently two times with equivalent results.

and associated potential toxicities[21]. Aqueous surfactant stabilized M1TFV, M2TFV and TAF nanocrystals (NM1TFV, NM2TFV, and NTAF) were therefore produced by high-pressure homogenization. Drug encapsulation efficiencies for NM1TFV, NM2TFV, and NTAF nanocrystals were 94.6, 92.2, and 78.8%, respectively. The size, polydispersity index (PDI), and zeta potential for NM1TFV, were $215 \pm 10.3$ nm, $0.30 \pm 0.03$, and $-16.4 \pm 0.6$ mV, respectively, and remained stable at 25 °C for at least 90 days (Fig. 2a). The size, polydispersity index (PDI), and zeta potential for NM2TFV, were $261.6 \pm 12.9$ nm, $0.38 \pm 0.08$, and $-22.7 \pm 0.7$ mV, respectively, and remained stable at 25 °C for at least 10 days (Fig. 2a). The size, PDI, and zeta potential for NTAF were $284.3 \pm 14$ nm, $0.3 \pm 0.005$, and $6.55 \pm 0.15$ mV, respectively, and remained stable at 25 °C for at least 30 days (Fig. 2a). We then determined the effect of the prodrugs on mitochondrial function by MTT in MDM. No adverse effects on mitochondrial activity or cell viability were observed at 200 μM in MDM for NM1TFV and NM2TFV after incubation with drug for 8 h (Fig. 2b). In contrast, a decrease in cell viability was observed for TAF at concentrations above 12.5 μM (Fig. 2b). We next evaluated the ProTides for antiretroviral activity. The calculated $EC_{50}$ values for unformulated M1TFV and M2TFV in human monocyte-derived macrophages (MDMs) were 109 nM and 45.75 nM, respectively (Fig. 2c). The observed potency is likely linked to poor prodrug solubility upon dilution and precipitation in aqueous culture media. Notably, nanoformulation of the prodrugs significantly improved drug potency, with calculated $EC_{50}$ values of 0.71 and 2.08 nM for NM1TFV and NM2TFV, respectively (Fig. 2c). The antiretroviral activity of TAF ($EC_{50}$ 0.22 nM) following nanoformulation into NTAF ($EC_{50}$ 0.44 nM,

Supplementary Fig. 9a) was equivalent. In humans, CD4+ T-cell line (CEM-ss CD4+ T-cells), the $EC_{50}$ of NM1TFV, NM2TFV, and TAF were 28.8, 30.0, and 0.33 nM respectively (Supplementary Fig. 9a, c).

**Cellular drug uptake and retention.** Delivery of solid drug nanoparticles to macrophages has been shown to improve the half-life of ART through the formation of intracellular and tissue drug storage sites[11,12]. Macrophage nanoparticle uptake and storage are facilitated by their phagocytic and proinflammatory functions, large cytoplasmic to the nuclear ratio for drug storage, and their significant role in viral infection makes them ideal targets for ART[16]. We, therefore, evaluated NM1TFV, NM2TFV, and TAF in MDMs at concentrations of 10 and 100 μM. It is worth noting that evaluation of TAF in MDMs was performed at 10 μM since 100 μM was found to be highly toxic to cells. Intracellular prodrug concentrations were evaluated over 8 h. Previous work from our laboratory has shown that maximal drug nanoparticle uptake by MDM occurs within 8 h[12,33,34]. As shown in Fig. 3a, b, the highest drug uptake was observed with NM1TFV. Intracellular prodrug concentration for NM1TFV at 8 h after treatment with 10 μM of drug was 8.62 μg/$10^6$ cells (Fig. 3a), a 4.2- and 22.7-fold higher concentration than that observed for NM2TFV (2.04 μg/$10^6$ cells) or TAF (0.38 μg/$10^6$ cells), respectively. For 100 μM drug treatment after 8 h (Fig. 3b), intracellular prodrug concentration for NM1TFV was 42.36 μg/$10^6$ cells, a 2.2-fold higher concentration than that observed for NM2TFV (19.4 μg/$10^6$ cells). Visualization of MDMs treated with equimolar drug nanoparticles by TEM revealed greater

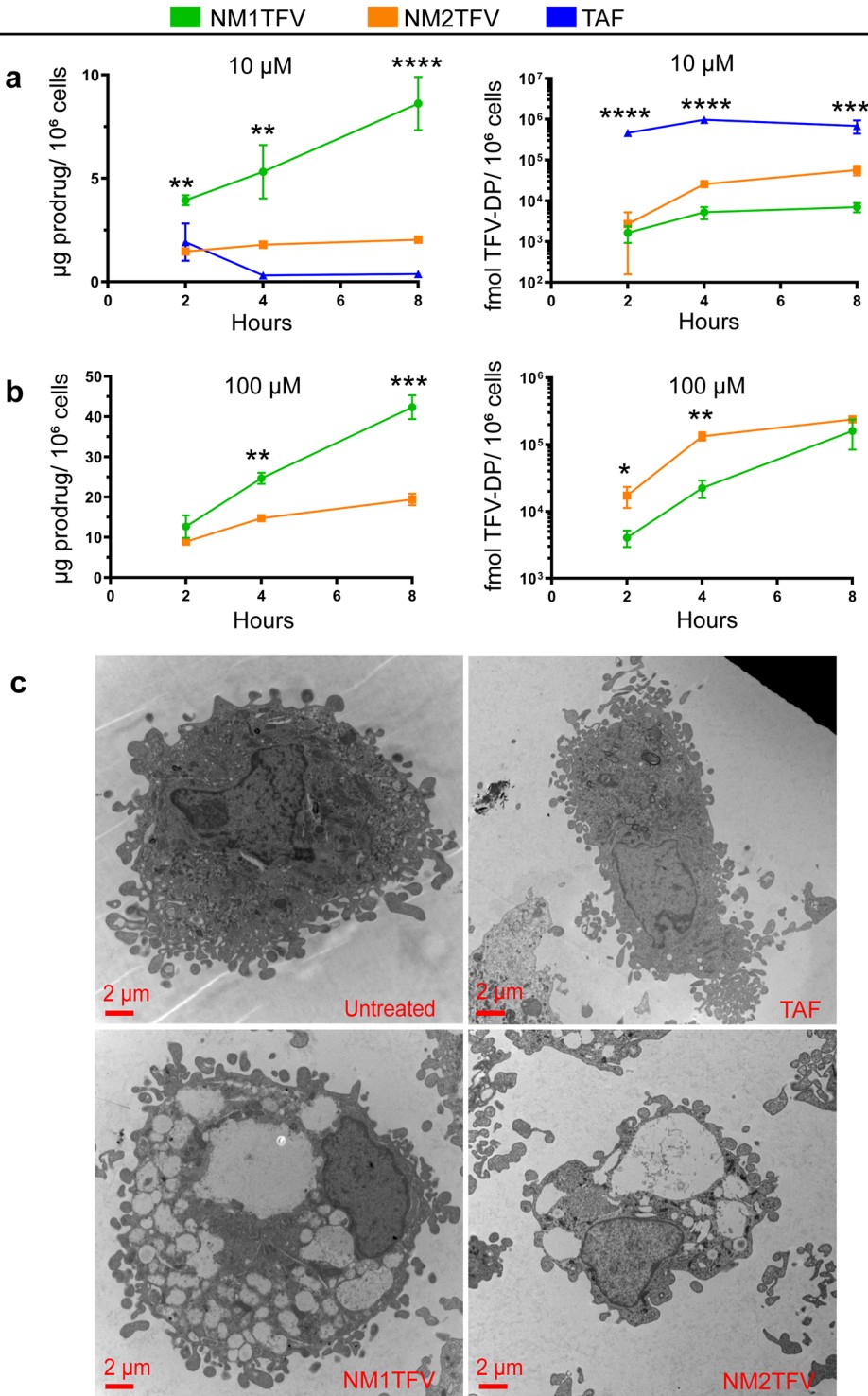

**Fig. 3 Cellular uptake.** Following **a** 10 µM (**P = 0.0035, **P = 0.0029, ****P < 0.0001 NM1TFV vs. NM2TFV prodrug; ****P < 0.0001 NM1TFV vs. TAF TFV-DP) or **b** 100 µM treatments (**P = 0.0021, ***P = 0.0003 NM1TFV vs. NM2TFV prodrug; *P = 0.0273, **P = 0.0022 NM1TFV vs. NM2TFV TFV-DP), intracellular prodrug (left) and TFV-DP (right) concentrations in MDM was measured over an 8 h period. **a, b** Prodrug uptake of NM1TFV (green) significantly greater than NM2TFV (orange) and TAF (blue). TAF treatment resulted in the most rapid prodrug conversion to TFV-DP, followed by NM2TFV, and NM1TFV. **a, b** Values reported are the mean ± SEM of three replicates. Each experiment was repeated independently three times with equivalent results. **c** Transmission electron microscopy (TEM) of control, TAF, NM1TFV, and NM2TFV loaded MDM after 8 h drug incubation with 10 µM drug. Scale bar: 2 µm. For comparison of two groups Student's t-test (two-tailed, unpaired) was used (*P ≤ 0.05, **P < 0.01, ***P < 0.001, ****P < 0.0001 NM1TFV compared with NM2TFV). A one-way analysis of variance (ANOVA) followed by Tukey's post hoc test was used to compare the prodrug and TFV-DP levels among three treatment groups (*P ≤ 0.05, **P < 0.01, ***P < 0.001, ****P < 0.0001 NM1TFV compared with NM2TFV or TAF). Data were independently reproduced three times.

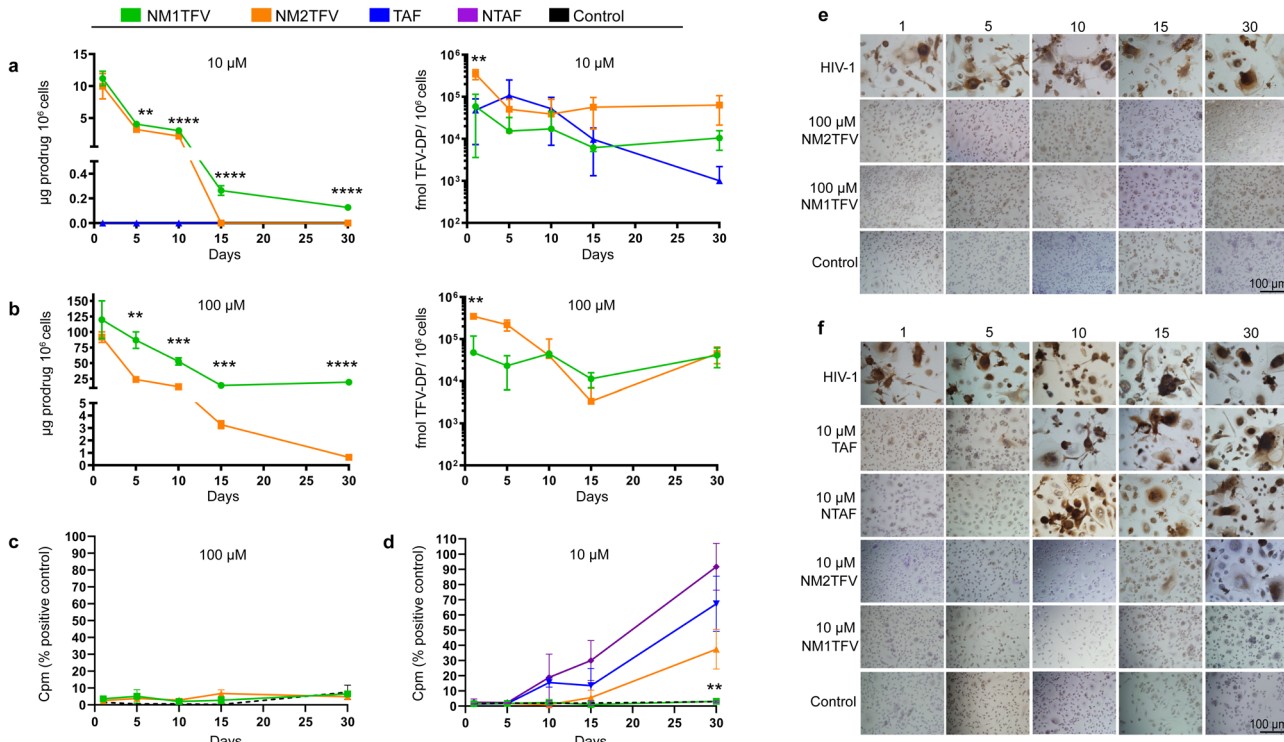

**Fig. 4 Long-term cell nanoparticle interactions.** Following **a** 10 µM (**P = 0.0026, ****P < 0.0001 NM1TFV vs. NM2TFV prodrug; **P = 0.0096 NM1TFV vs. NM2TFV TFV-DP) or **b** 100 µM treatments of NM1TFV (green), NM2TFV (orange) or TAF (blue) (**P = 0.001511, ***P = 0.000326, ***P = 0.000180, ****P < 0.0001 NM1TFV vs. NM2TFV prodrug on days 5, 10, 15, and 30, respectively; **P = 0.001911 NM1TFV vs. NM2TFV TFV-DP), intracellular prodrug (left) and TFV-DP (right) concentrations in MDM were measured over a 30-day period. Drug concentrations are expressed as mean ± SEM, N = 3 biological replicates. Following **c**, **e** 100 µM or **d**, **f** 10 µM treatments, the antiretroviral activities of NM1TFV and NM2TFV were determined by measures of HIV-1 RT activity in culture supernatants and by cell-associated HIV-1 p24 antigens by immunonocytochemistry. **P = 0.0042 NM1TFV vs. NM2TFV. **e**, **f** Scale bar: 100 µm. HIV-1 RT activity is expressed as mean ± SEM, N = 4 biological replicates. For comparison of two groups Student's t-test (two-tailed, unpaired) was used (*P ≤ 0.05, **P < 0.01, ***P < 0.001, ****P < 0.0001 NM1TFV compared with NM2TFV). A one-way analysis of variance (ANOVA) followed by Tukey's post hoc test was used to compare the prodrug and TFV-DP levels among three treatment groups (*P ≤ 0.05, **P < 0.01, ***P < 0.001, ****P < 0.0001 NM1TFV compared with NM2TFV).

accumulation of NM1TFV and NM2TFV drug reservoirs in the cytoplasm compared to TAF (Fig. 3c). In CEM-ss T-cells, significantly improved prodrug uptake was observed for NM1TFV compared to NM2TFV and NTAF (Supplementary Fig. 9b). To further confirm the release of ProTides from the nanoparticles and their conversion to active metabolites, we quantified intracellular TFV-DP levels in MDMs after a single exposure to NM1TFV, NM2TFV, or TAF. As shown in Fig. 3a, the 10 µM TAF treatment produced the highest TFV-DP concentration, peaking at 8 h (690,949 fmol/10^6 cells). For NM1TFV at 10 µM, the maximum TFV-DP levels were observed at 8 h (7,056.7 fmol/10^6 cells), and of 57,113 fmol/10^6 cells at 8 h for NM2TFV. Even though lower concentrations of TFV-DP were measured for NM1TFV compared to NM2TFV or TAF, the active metabolite levels were sustained over 8hrs, well above the human EC$_{90}$ of 40 fmol/10^6 cells. At the 100 µM concentration (Fig. 3b), the TFV-DP concentrations for NM1TFV and NM2TFV peaked after 8 h, and the difference was not found to be statistically significant (160,867 fmol/10^6 cells, and 239,422 fmol/10^6 cells, respectively).

Prior data from our laboratories demonstrate sustained and enhanced storage of nanoformulations in MDM[35]. Thus, retention of TFV nanoformulations in MDM was determined. Following a 10 µM treatment, NM1TFV facilitated retention of the prodrug in MDM for up to 30 days (0.13 µg/10^6 cells), while NM2TFV and native TAF treatments showed no detectable drug within 15 days, and 24 h respectively (Fig. 4a). At a higher concentration of 100 µM (Fig. 4b) both NM1TFV and NM2TFV

facilitated prodrug retention for up to 30 days (19.53 µg/10^6 cells and 0.64 µg/10^6 cells respectively). To assess the extent of active metabolite retention, intracellular TFV-DP was measured over a 30-day time period after a single exposure of drug for 8 h. Notably, a single treatment of MDMs with NM1TFV or NM2TFV provided steady and sustained TFV-DP levels over the entire 30-day period (Fig. 4a, b). As shown in Fig. 4a, the amount of TFV-DP retained by 10 µM NM1TFV-treated MDM was 48,232.4 fmol/10^6 cells on day 1, and 10,507.7 fmol/10^6 cells on day 30. Similarly, 10 µM of NM2TFV provided sustained TFV-DP concentrations of 349,162 and 64,057.4 fmol/10^6 cells at days 1 and 30, respectively (Fig. 4a). TAF demonstrated a rapid decline of the active metabolite concentration compared to NM1TFV or NM2TFV; levels of TFV-DP fell from 59,239.4 fmol/10^6 cells at day 1 to 1,013.4 fmol/10^6 cells on day 30. Sustained intracellular TFV-DP concentrations were also observed at the 100 µM of NM1TFV or NM2TFV treatments. For NM1TFV, intracellular active metabolite TFV-DP was retained at levels of 47,622.4 fmol/10^6 cells at day 1 then 41,219.4 fmol/10^6 cells by day 30. Similarly, NM2TFV provided sustained TFV-DP concentrations of 348,132 and 45,144.8 fmol/10^6 cells at days 1 and 30, respectively (Fig. 4b). Such sustained-release formulations could maintain effective drug concentrations at cellular and tissue reservoirs of infection. To assess whether enhanced intracellular TFV-DP levels would translate into improved efficacy, MDM were challenged every five days with HIV-1$_{ADA}$ to an end point of 30 days following an 8 h drug loading and assayed quantitatively

for HIV-1 RT activity, as well as qualitatively for HIV-1 p24 antigen expression (Fig. 4c−f). At the 100 μM dose, no statistically significant differences in RT activity were detected in media from NM1TFV and NM2TFV-treated cells (Fig. 4c). At 10 μM, significantly lower RT activity was detected in media from NM1TFV and NM2TFV-treated cells compared to TAF and NTAF treatment beginning on day 10 post-treatment and continuing out to 30 days (Fig. 4d). NM1TFV treatment also resulted in significantly lower RT activity compared to NM2TFV at day 30 (Fig. 4c, d). Both 10 and 100 μM NM1TFV suppressed HIV-1 RT activity by >92% and >96% at all time points, respectively (Fig. 4c, d). While 100 μM NM2TFV also suppressed HIV-1 RT activity by >92% at all the time points, treatment with 10 μM of NM2TFV provided viral inhibition of 62.57% at day 30. Similarly, 10 μM NTAF and TAF provided viral inhibition of 81.1 and 84.5% respectively at day 10, with complete viral breakthrough occurring at day 30 (Fig. 4d, f). In all, these findings demonstrate that NM1TFV improves intracellular delivery of TFV-DP with enhanced drug potency and sustained antiretroviral efficacy.

**PK testing**. To assess the PK and drug biodistribution (BD) profiles, male and female Sprague-Dawley rats were injected intramuscularly (IM) with a single dose of 75 mg/kg body weight TFV-equivalents of NM1TFV, NM2TFV, or NTAF. The area under the curve (AUC) for whole blood prodrug (6,427 (ng* days)/mL) in NM1TFV treated animals was found to be statistically higher than whole blood prodrug (2,792 (ng* days)/mL) in NM2TFV animals (GraphPad Prism v9.0.0.0 software, one-way ANOVA, $P = 0.0340$) (Fig. 5a). Additionally, compared to NM2TFV, NM1TFV resulted in significantly greater whole blood prodrug concentrations on days 14 and 42 (Fig. 5a). Notably, NM1TFV treatment resulted in significantly higher TFV concentrations in whole blood compared to NTAF on days 14, 28, and 56 ($p < 0.0001$, $p = 0.0326$, $p = 0.024$, respectively). However, no statistically significant differences in whole blood TFV levels were observed between NM1TFV and NM2TFV treatment groups (Fig. 5a). NM1TFV significantly improved tissue drug delivery compared to NM2TFV and NTAF. Specifically, compared to NTAF and NM2TFV, NM1TFV provided significantly higher TFV concentrations in spleen, lymph nodes, liver, gut, kidneys, and site of injection at 28 and 56 days-post drug administration (Fig. 5b–f and Supplementary Fig. 10). For NM1TFV at day 56, the TFV levels were 102.4, 26, 12.5, 12.5, 1,258.1, and 8,883.2 ng/g tissue in the liver, spleen, lymph nodes, gut, kidney, and site of injection, respectively. By comparison, TFV levels 56 days after NTAF treatment were 0.1, 1.8, 1.03, 5.58, and 18.8 ng/g tissue in liver, lymph nodes, gut, kidneys, and site of injection, respectively. Similarly, TFV levels at day 56 after NM2TFV treatment were 4.87, 1.58, 0.1, 20.28, and 193.98 ng/g tissue in the liver, spleen, gut, kidney, and site of injection, respectively. Also, NM1TFV displayed substantially higher prodrug levels compared to other treatments (Fig. 5b–f and Supplementary Fig. 10). For NM1TFV at day 56, the prodrug levels were 65.6, 151.3, 27.8, 18.3, 5.3, and 433,475 ng/g tissue in the liver, spleen, lymph nodes, gut, kidney, and site of injection, respectively (Fig. 5b−f, Supplementary Fig. 10b). The data supports the contention that NM1TFV undergoes slow nanoparticle dissolution and prodrug hydrolysis in vivo compared to NM2TFV and NTAF. At the site of injection, significantly greater levels of prodrug were observed for NM1TFV at day 28 (1,253,500 ng/g tissue) and 56 (433,475 ng/g tissue) (Fig. 5f). Based on the initial dose, the total amount of drug at the site of injection after NM1TFV injection was 10% at day 28 and 5% at day 56 (Supplementary Fig. 11b). For NM2TFV, the muscle drug

depot was 0.1% and 0.05% of the initial dose at days 28 and 56 (Supplementary Fig. 11b). The molar ratios of the prodrug to TFV levels in the muscle demonstrated that a significant proportion of drug after NM1TFV treatment remained in prodrug form when compared against NM2TFV (Supplementary Fig. 11c). Additionally, Pearson correlation determined that prodrug concentration at the site of an injection directly correlated with prodrug and parent drug concentration in most of the tissues, supporting a linear relationship between the magnitude of the prodrug intramuscular depot and tissue drug concentrations (two-tailed, α = 0.05, Supplementary Fig. 11d). In all, the data suggest that NM1TFV is more stable at the site of injection and is absorbed more slowly than NM2TFV from the intramuscular depot. Given the enhanced stability of the M1TFV prodrug in the muscle at day 56, NM1TFV offers the potential for sustained drug delivery beyond two months.

**TFV-DP measurements**. We next determined the concentration of the active metabolite in key HIV and hepatitis B virus (HBV) target cells and tissues (Fig. 6). In PBMCs, a single treatment with either NM1TFV, NM2TFV, or NTAF demonstrated TFV-DP concentrations above the human $EC_{90}$ (40 fmol/$10^6$ cells) during the entire study duration (Fig. 6a). Specifically, TFV-DP levels for NM1TFV, NM2TFV, and NTAF at 8 weeks were 156.28, 291.75, and 305.33 fmol/$10^6$ cells, respectively. Similarly, no significant differences in TFV-DP levels were recorded in splenoctyes, non-parenchymal liver cells, and in vaginal tissue among the three treatments (Fig. 6). However, NM2TFV resulted in significantly higher TFV-DP levels in lymphocytes at D28 (239.319 fmol/$10^6$ cells), compared to NM1TFV (156.26 fmol/$10^6$ cells) and NTAF (71.94 fmol/$10^6$ cells) (Fig. 6b). Compared to NM2TFV and NTAF, NM1TFV provided significantly higher TFV-DP levels in parenchymal liver cells at D56 (328.02 fmol/$10^6$ cells) and in rectal tissue at all time points (36,575.61 fmol/$10^6$ cells on D28, and 11,276.31 fmol/$10^6$ cells on D56, Fig. 6d, f). Importantly, at the end of the study, NM1TFV exhibited sustained PBMC and tissue TFV-DP concentrations above the level associated with effective pre-exposure prophylaxis in humans. Given the high muscle M1TFV levels at day 56, it is feasible to achieve sustained therapeutic TFV-DP levels beyond 2 months.

**Toxicology and evaluation of injection site**. Toxicity was assessed in SD rats after NM1TFV, NM2TFV, and NTAF treatment. Animal weights were recorded weekly and organ to body weight ratios were determined on the days of sacrifice (days 28 and 56); plasma and tissues were collected for hematologic, chemistry, and histopathology (Fig. 7 and Supplementary Figs. 12−14). Controls were age and sex-matched and evaluated in replicate untreated rats. Erythema and edema were not observed at the site of injection in any of the treatment groups. No statistically significant differences in body weights were observed between the treatment groups (Supplementary Fig. 13b). Transient differences in lung and spleen to body weight ratios were observed between NM1TFV and NTAF groups at day 28. However, no statistically significant differences in organ to body weight ratios were detected on day 56 (Supplementary Fig. 13a). No hematological abnormalities were detected (Supplementary Fig. 13c). There were no notable differences between treatment groups for comprehensive chemistry profiles (Supplementary Fig. 12), suggesting that NM1-, NM2- TFV and NTAF did not adversely affect organ function. Hematoxylin and eosin-stained tissue sections, examined by a certified pathologist in a blinded fashion, revealed no abnormal pathology in the kidney and livers of NM1TFV and NTAF treated animals (Supplementary Fig. 14b). Injection site reactions were examined following

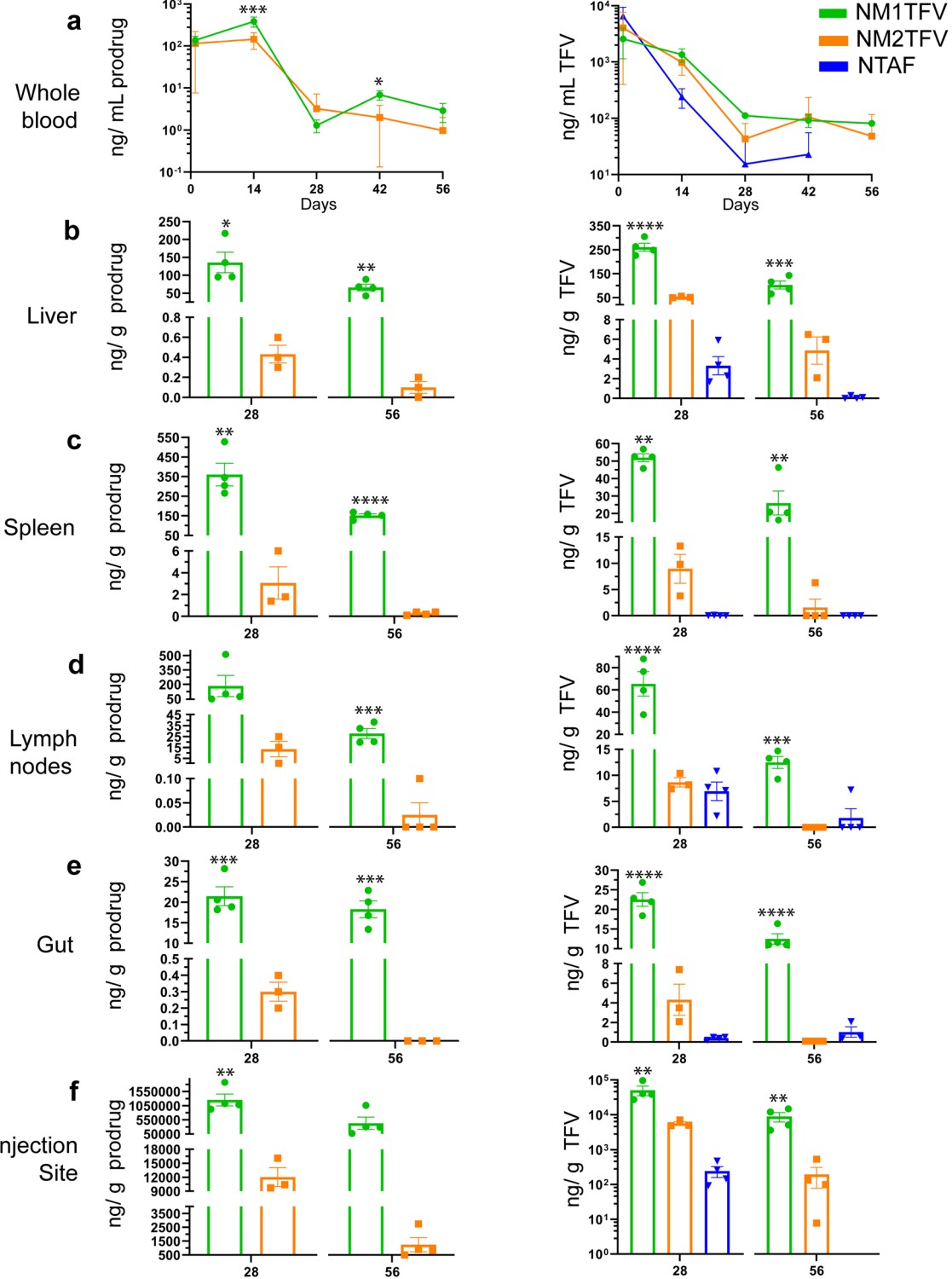

**Fig. 5 Pharmacokinetics.** SD rats were administered a single IM dose of NM1TFV (green), NM2TFV (orange), or NTAF (blue) (75 mg/kg TFV-eq.) to determine pharmacokinetic (PK) profiles. **a** Prodrug (left) and TFV (right) concentrations in blood were determined at days 1 and 14 then every two weeks for 8 weeks (***$P = 0.0001$, *$P = 0.0177$ NM1TFV vs. NM2TFV prodrug). Tissue biodistribution of M1TFV, M2TFV, and TFV was assessed at 28 and 56 days after injection in the **b** liver (*$P = 0.0107$, **$P = 0.002$ NM1TFV vs. NM2TFV prodrug; ****$P < 0.0001$, ***$P = 0.0007$ NM1TFV vs. NM2TFV TFV), **c** spleen (**$P = 0.0034$, ****$P < 0.0001$ NM1TFV vs. NM2TFV prodrug; **$P = 0.004$, **$P = 0.0055$, NM1TFV vs. NM2TFV TFV), **d** lymph node (***$P = 0.0009$ NM1TFV vs. NM2TFV prodrug; ***$P = 0.0001$, ****$P < 0.0001$ NM1TFV vs. NM2TFV TFV), **e** gut (***$P = 0.0006$, ***$P = 0.0007$ NM1TFV vs. NM2TFV prodrug; ****$P < 0.0001$ NM1TFV vs. NM2TFV TFV), and **f** site of injection (**$P = 0.0043$ NM1TFV vs. NM2TFV prodrug; **$P = 0.0084$, **$P = 0.0085$ NM1TFV vs. NM2TFV TFV). Data are expressed as mean ± SEM where $N = 4$ biological replicates. Prodrug concentrations were compared using two-tailed unpaired *t*-test (****$P < .0001$, ***$P < .001$, **$P < .01$, *$P < .05$ NM1TFV compared with NM2TFV). To compare TFV levels between three or more groups, ordinary one-way ANOVA followed by Tukey's post hoc test with a single pooled variance was used (****$P < .0001$, ***$P < .001$, **$P < .01$, *$P < .05$ NM1TFV compared with NM2TFV).

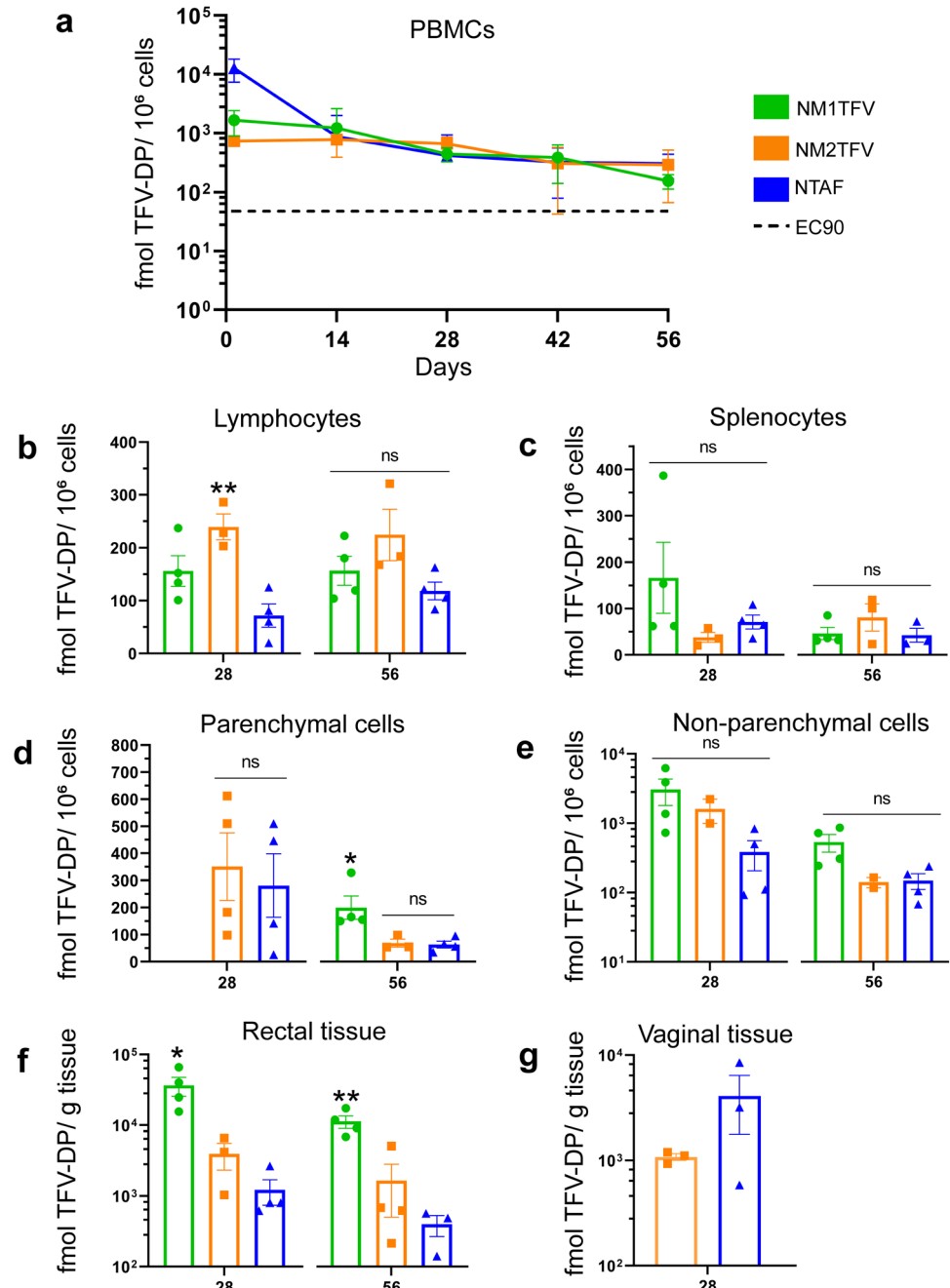

**Fig. 6 In vivo TFV-DP Measurements.** SD rats were administered a single IM dose of NM1TFV (green), NM2TFV (orange), or NTAF (blue) (75 mg/kg TFV-eq.) to determine pharmacokinetic (PK) profiles. **a** TFV-DP concentrations in PBMCs were determined at days 1 and 14 then every two weeks for 8 weeks. Tissue biodistribution of TFV-DP was assessed at 28 and 56 days after injection in the **b** lymphocytes (**P = 0.0024, NM2TFV vs. NTAF), **c** splenocytes, **d** parenchymal liver cells (*P = 0.0372, NM1TFV vs. NM2TFV), **e** non-parenchymal liver cells, **f** rectal tissue (*P = 0.0115, **P = 0.0058 NM1TFV vs. NM2TFV), and **g** vaginal tissue. Data are expressed as mean ± SEM where N = 4 biological replicates. Vaginal TFV-DP concentrations were compared using a two-tailed unpaired t-test (****P < .0001, ***P < .001, **P < .01, *P < .05 NM1TFV compared with NM2TFV). **a**−**f** To compare TFV-DP levels between three or more groups, ordinary one-way ANOVA followed by Tukey's post hoc test with a single pooled variance (****P < .0001, ***P < .001, **P < .01, *P < .05 NM1TFV compared with NM2TFV).

NM1TFV and NTAF injections showing minimal granulomatous reactions (Fig. 7 and Supplementary Fig. 14a). Specifically, one week after NM1TFV administration, amorphous material, a central area containing foamy macrophages and cellular debris were present at the injection site, typical of necrosis. Spindle-shaped fibroblasts and vascular proliferation surrounded the necrotic region, indicating wound healing and granulation tissue formation. Histiocytic infiltration of the intramuscular depot was

also observed, including minimal infiltration of the adjacent skeletal muscle. Regions of extracellular amorphous eosinophilic material were also noted. Interspersed within the nuclear debris and foamy histiocytes were small, spherical basophilic granules, mostly extracellular. Given that NM1TFV carries a negative surface charge (Fig. 2a) and is readily stained by hematoxylin (Fig. 7a), the observed basophilic granules were recorded as NM1TFV nanoparticle aggregates. Clusters of these basophilic

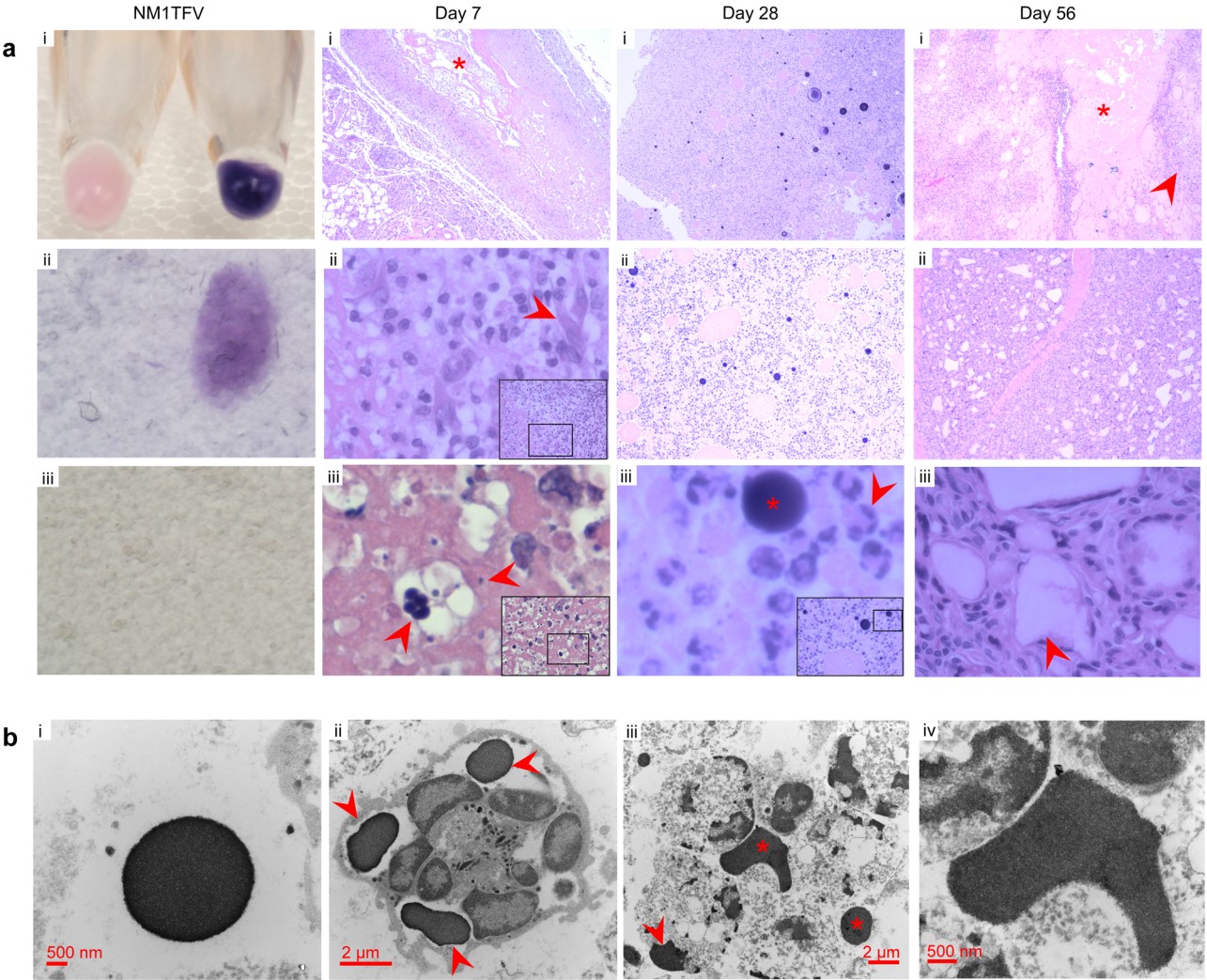

**Fig. 7 Injection site histopathology.** Following a single IM dose of NM1TFV or NTAF male SD rats were sacrificed on days 7, 28, and 56. **a** H&E staining of NM1TFV (i) revealed the formulation was readily stained by hematoxylin (ii), but not eosin (iii). For the histological examination of the site of injection, 5 µM sections of paraffin-embedded tissues were stained with hematoxylin and eosin. Images were captured using a Nuance EX multispectral imaging system affixed to a Nikon Eclipse E800 microscope (Nikon Instruments). At day 7, a necrotic focus was observed (i, indicated by *), surrounded by histiocytes, and fibroblasts (ii, arrow). Clusters of foreign material, likely NM1TFV were also observed interspersed in nuclear debris (iii, arrows). By day 28, larger extracellular basophilic staining globules (iii, indicated by *) along with small basophilic intracellular granules present in the histiocytes (iii, indicated by arrow), along with foreign body granulomatous reaction. At 56 days, the foreign material appeared mostly amorphous (i, indicated by *), with a foreign body giant cell granulomatous reaction (ii, iii). **b** TEM images of the NM1TFV site of injection at day 28 (i & ii), and 56 (iii & iv) confirmed the light microscopic findings. **a, b** Study was performed once; images are representative of the histological findings from three biological replicates. Magnification of each micrograph is as follows: 1×, 60×, 60× for NM1TFV i, ii, iii, respectively. 4×, 20×, 40× for Day 7 i, ii, iii, respectively. 4×, 10×, 40× for Day 28 i, ii, iii, respectively. 4×, 4×, 40× for Day 56 i, ii, iii, respectively.

granules were noted 7 days following the intramuscular injection of the prodrug nanoformulation, suggesting aggregation of NM1TFV particles (Fig. 7a). By day 28, there were large extracellular basophilic staining globules along with small basophilic intracellular granules present in the histiocytes, indicating further aggregation of NM1TFV (Fig. 7a). This was accompanied by a foreign body granulomatous reaction, including foreign body giant cells (Fig. 7a). When sections were imaged by TEM, the basophilic granules observed by light microscopy appeared as non-cellular electron-dense aggregates, further suggesting that this foreign material was the NM1TFV formulation and corresponding to the presence of the prodrug by chemical analysis. These electron-dense aggregates of NM1TFV appeared extracellularly and intracellularly within the histiocytes (Fig. 7b). At

56 days, the foreign material appeared mostly amorphous, with a foreign body giant cell granulomatous reaction. Extracellular basophilic globules and finer granular intracellular basophilic material were present, indicating sustained phagocytosis of NM1TFV by histiocytes (Fig. 7). One rat at 56 days showed a small focus of fibrosis. TEM observations supported the light microscopic findings, showing extracellular electron-dense aggregates of NM1TFV along with intracellular aggregates within the histiocytes (Fig. 7b). The injection sites in rats treated with NTAF showed small areas of histiocytic infiltrate at 28 days, mostly in the interstitium (Supplementary Fig. 14a). Amorphous foreign material was present, and this persisted in one of the three rats at 56 days, including a foreign body giant cell granulomatous reaction (Supplementary Fig. 14a). There was no evidence of

muscle degeneration, basophils or eosinophils, cellulitis, or abscesses in any of the treatment groups at any time point examined.

## Discussion

Although LA ARV formulations could impact HIV treatment and prevention, the physicochemical properties of current oral ARVs are not compatible with existing sustained release formulation approaches. Thus, improvements in drug regimens are emergently needed for PrEP and maintenance therapies. While current HIV treatment consists of three-drug regimens, and current PrEP consists of a two-drug regimen; both require once-daily dosing. Change in these paradigms is now possible. This resides in harnessing the potential role of TAF in LA delivery systems. Indeed, TAF and TDF prodrugs demonstrate potent activity against HIV and HBV infections making the agent the most used antiretroviral[36,37]. The combination of TFV and emtricitabine is effective at preventing HIV-1 transmission when taken daily[38,39]. Additionally, the current recommended first-line antiretroviral regimen for HIV treatment consists of combinations of TAF/TDF and emtricitabine together with an integrase, protease, or non-nucleoside reverse transcriptase inhibitor[40]. Despite its effectiveness, the need for strict regimen adherence, noted toxicities, and restricted tissue penetrance all highlight immediate needs for improved formulation strategies[41–43]. Thus, LASER ART and delivery devices for ARVs are being developed to extend dosing intervals, reduce systemic toxicity, and improve PK profiles[4,16,21,44–46]. Herein, we report on the development of TFV ProTides with improved prodrug stability. In this approach, the ionizable phosphate groups in the drug are masked by aromatic and amino acid ester moieties, which upon delivery undergo intracellular enzymatic or chemical hydrolysis to release the free form of the phosphorylated compounds for subsequent activation and competitive inhibition of the viral reverse transcriptase. The lead TFV ProTide candidate (M1TFV) facilitated the production of stable aqueous nanocrystals that exhibited enhanced drug and active metabolite delivery into macrophages and CEM-ss CD4+ T cells. PK and efficacy evaluations showed significant improvements in drug apparent half-life, biodistribution, and antiretroviral activities over TAF. These data, taken together, support the idea that NM1TFV could enable sustained release of TFV-DP to enable dosing intervals for TFV beyond every two months.

TAF is a ProTide of the acyclic nucleoside phosphonate drug tenofovir. The application of ProTide technology to TFV led to the discovery of TAF that is characterized by enhanced potency, improved lymphoid tissue drug delivery, and has also limited the risk for nephrotoxicity and loss of bone mineral density associated with another prodrug of tenofovir, tenofovir disoproxil fumarate (TDF)[18,19]. Despite the advantages conferred by TAF, its application in LA systems remains limited by the inherent hydrolytic instability of the prodrug. The enhanced potency and chemical composition of TAF further limit the possibility of making direct chemical modifications to the prodrug to improve compound stability without affecting efficacy and safety profiles. Moreover, prior attempts to improve prolonged TAF stability by either using excipients, formulation approaches, or pH modulation are yet to demonstrate successful in vivo outcomes[22,45]. To these ends, we explored TFV ProTides bearing safe but bulky hydrophobic lipids not explored by conventional ProTide designs due to their presumed undesired effect on compound dissolution properties in physiological fluids to affect efficacy. Two hydrophobic and lipophilic stable docosyl phenylalanyl and alanyl ester TFV ProTides (M1TFV, M2TFV) were created and nanoformulated by high-pressure homogenization as aqueous suspensions of crystalline drugs stabilized by P407 surfactant. Even though

solubility and dissolution rates are inherent properties of organic compounds, the actual dissolution rate is dependent on the surface area and degree of saturation. The reduced particle size was expected to enhance prodrug dissolution to produce rapid absorption and therapeutic onset upon administration. Since solubilized form of TAF is pH and water-sensitive, saturated aqueous solid drug nanoparticles were prepared at a pH of 5.5 to maintain prodrug stability within the formulation. The recorded uniform particle sizes, narrow polydispersity indices, and negative zeta potentials for NM1TFV and NM2TFV formulations are suggestive of nanoparticle stability and homogeneity. Importantly, the high drug loading capacity for the TFV ProTide nanocrystals could facilitate reduced injection volumes.

After producing and characterizing the stable TFV ProTide formulations, we then evaluated their abilities to enter and slowly release the drug within HIV-1 target cells. Nanocrystal delivery to phagocytic and highly mobile macrophages facilitates drug transfer to CD4+ T cells and other restricted tissue reservoirs of infection[12]. We have previously demonstrated that once inside macrophages, drug nanocrystals are stored in the late- and recycling-endosomes[35]. This intracellular particle storage protects drugs from systemic metabolism and prolongs half-life, ultimately leading to improved PK and drug BD profiles[11,12]. Also, the transformation of ProTides into active nucleotide metabolites to competitively inhibit the activity of HIV reverse transcriptase is mediated by intracellular kinases, underscoring the importance of cellular NRTI delivery. Compared to NTAF and NM2TFV, treatment of human MDM with NM1TFV resulted in enhanced cellular uptake and intracytoplasmic retention of the prodrug and TFV-DP over 30 days. The observed intracellular TFV-DP levels for NTAF and NM2TFV were expected given their rapid hydrolysis rates when compared to NM1TFV. Prior studies have also shown that TFV-DP exhibits an intracellular half-life of 60–100 h[4] and is likely to be extended when delivered in form of solid drug nanoparticles that readily undergo dissolution prior to prodrug release, hydrolysis, and activation. Consequently, NM1TFV exhibited sustained antiretroviral activity as determined by HIV-1p24 antigen expression and RT activity. Together, these characterization data sets demonstrate that NM1TFV improves prodrug stability, cell uptake, retention, and efficacy.

Optimal approaches for effective treatment and prevention of HIV-1 infection should deliver and sustain therapeutic drug concentrations at cellular and tissue sites of infection[47]. We therefore investigated the abilities of NTAF, NM1TFV, and NM2TFV to improve drug PK and biodistribution. Pharmacokinetic tests in SD rats demonstrated TFV-DP levels in PBMCs at or above 100 fmol/million cells for 56 days for all treatments. Notably, NM1TFV exhibited sustained intramuscular and enhanced lymphatic tissue drug levels compared to NM2TFV and NTAF. Toxicological assessments (CBC, comprehensive serum chemistry profiles, organ to body weight ratios, kidney and liver histopathology) demonstrated that neither NM1TFV, NM2TFV, nor NTAF induced systemic or organ-specific adverse effects. Following intramuscular administration of nanoformulations, the injection site serves as a primary depot from where the drug is slowly absorbed and eventually released into circulation (Figs. 5, 7 and Supplementary Fig. 10). Though, defining the precise mechanisms of distribution of the intramuscular drug nanosuspensions represents a current knowledge gap. Prior studies have shown that absorption of solid drug nanocrystals following intramuscular administration is dependent on particle size and rate of dissolution, host protein adsorption on the particle surface, and type of primary depot formed (particle agglomerates or dispersion)[48]. In these prior studies, the local inflammatory response was found to affect nanoparticle phagocytosis efficiency, prodrug stability, and uptake of lipophilic compounds by the

lymphatic system[48–52]. For TFV prodrugs, minimizing extracellular prodrug hydrolysis at the site of injection would limit the accumulation of polar and ionic TFV or ProTide degradation products in the muscle. Compared to TAF prodrug, earlier studies have demonstrated direct links between extracellular TFV and delayed wound healing, marked alterations in epidermal cell proteomics and ulcerations[53,54], underscoring the importance of prodrug stability in extracellular compartments. Of particular note, NM1TFV demonstrated sustained prodrug stability at the site of injection.

Intramuscular injection of NM1TFV nanocrystals induced typical, expected subclinical granulomatous foreign body reactions[55]. The observed granuloma formation and subsequent intracellular sequestration of NM1TFV by infiltrating histiocytes and giant cells facilitate drug uptake and accumulation within reticuloendothelial and lymphoid tissues to create secondary drug depots (Figs. 5, 6e, 7). Inflammatory granulomatous tissue reactions have previously been observed for similar FDA-approved long−long-acting injectable aqueous drug nanosuspesions[48–52] where macrophage infiltration and subsequent phagocytosis of nanocrystals is known to modulate biphasic flip-flop pharmacokinetics by promoting both prodrug dissolution and activation[48,50]. Similarly, injection site analysis by TEM demonstrated that the reticuloendothelial system and immune cell infiltration play important roles in lipophilic drug nanocrystal distribution into tissues[56]. Therefore, in addition to the inherent physicochemical properties of the active pharmaceutical ingredient and nanoparticle, local tissue responses likely play a considerable role in the enhanced drug biodistribution to the reticuloendothelial system and the sustained release profile of NM1TFV. Importantly, intramuscularly delivered NM1TFV did not exhibit severe injection site reactions such as allergic reactions, cellulitis, abscesses, or chronic necrosis, the latter of which has been associated with subcutaneous TAF implants[57]. In addition to improved PK and safety profiles, in proof-of-concept pharmacodynamics studies NM1TFV demonstrated antiviral efficacy in HBV-infected humanized mice[58]. These results are a first step towards producing a long-acting TFV for human use by affecting cell and tissue drug penetration, and antiretroviral activity, and drug apparent half-life.

Long-acting NM1TFV can potentially extend antiretroviral drug administration from daily to beyond bimonthly. The TFV ProTides reported in this work confer therapeutic and safety benefits over conventional ProTides or other previously published long-acting TFV formulations. For instance, a previous report demonstrated that TFV-DP levels in PBMCs of humanized mice declined below 100 fmol/$10^6$ cells less than a week after subcutaneous administration of 200 mg/kg TAF loaded nanoparticles[25]. In contrast, our ProTide formulations sustained TFV-DP levels above this level in PBMCs for 2 months using a dose 2.7 times lower than what was previously reported[25]. Therefore, these developed formulations may facilitate the reduction of injection volumes without compromising therapeutic efficacy. Another technology being explored for sustained release of TFV is implants[21,44–46]. Implantable devices containing TAF are of increased interest due to the drug's potency and long half-life of the active metabolite[22,59]. However, a recent report on subcutaneous TAF implants inserted in rabbits and rhesus macaques demonstrated substantial histological changes in tissues surrounding the implant site[57]. Significant tissue necrosis was noted after 5 weeks and worsened by 12 weeks in animals implanted with TAF-containing devices[57]. Though a necrotic focus was observed one week after prodrug formulation injection, in contrast to what had been reported for TAF implants, no adverse histological changes were observed at the site of injection for the 2-month time period of the study. This observation is linked, in part, to the sustained prodrug and formulation stability within the muscle. With injectables, one potential drawback includes the inability to remove the drug in the case of adverse events. Utilizing an oral lead to preemptively identify adverse drug reactions can minimize the risk of such events. Additionally, in the case of NM1TFV, the predominant drug depot at the site of injection could offer the possibility of surgical removal in emergent cases. Thus, NM1TFV could potentially overcome major obstacles associated with the transformation of TFV into long-acting products.

In the present study, NM1TFV significantly improved drug and TFV-DP biodistribution compared to NM2TFV and NTAF. These results underscore the importance of extended PK evaluations beyond two months to elucidate potential clinical dosing intervals. To this end, future studies will involve 6-month PK studies of NM1TFV in rats and rhesus macaques. Given recent reports from HIV Prevention Trials Network (HPTN) 083 and HPTN 084 studies demonstrating that long-acting injectable cabotegravir (CAB-LA) administered every two months is more effective than daily pills in preventing HIV acquisition[60], it is clear that long-acting formulations may represent an important tool in controlling the HIV epidemic. Given the important role of TFV prodrugs in combination therapies, the lead candidate NM1TFV could have a major impact on HIV treatment and prevention. TFV also inhibits major co-infections, such as HBV[61]. In vivo proof of concept studies to assess the efficacy of NM1TFV has demonstrated the ability of the formulation to suppress HBV replication in humanized mice over 4 weeks following a single intramuscular dose[58]. Future studies will examine the use of NM1TFV for both monotherapy for HBV infection, and for treatment of HIV−HBV coinfections.

In summary, two lipophilic aqueous stable TFV ProTide nanoformulations, NM1TFV and NM2TFV, were created to enhance intracellular drug delivery and retention. The lead NM1TFV formulation demonstrates enhanced and sustained PBMC and tissue TFV-DP levels four times higher than the $EC_{90}$ for two months after a single intramuscular drug administration into Sprague Dawley rats. NM1TFV provides an attractive candidate for beyond every two months dosing of TFV to improve treatment outcomes.

## Methods

Monophenyl tenofovir was purchased from BOC Sciences (Shirley, NY). TAF hemifumarate was a generous gift from Gilead Sciences Inc. (Foster City, CA). N-(carbobenzyloxy)- L-phenylalanine, N-(carbobenzyloxy)- L-alanine, dicholoromethane (DCM), diethyl ether, anhydrous methanol (MeOH), anhydrous chloroform, anhydrous tetrahydrofuran (THF), anhydrous acetonitrile (ACN), anhydrous N,N-dimethyl formamide (DMF), 1-octanol, imidazole, triethylamine, 1-docosanol, triethyl silane (Et$_3$Si), thionyl chloride, pluronic F127 (P407), polyethylene glycol 3350, ciprofloxacin, 3-(4,5-dimethylthiazol-2-yl)-2,5-diphenyltetrazolium bromide (MTT), dimethyl sulfoxide (DMSO) ammonium bicarbonate, HEPES buffer solution, potassium chloride, ciprofloxacin, paraformaldehyde (PFA) and 3,3′-diaminobenzidine were purchased from Sigma-Aldrich (St. Louis, MO). Palladium, 10% on activated carbon, was purchased from STREM Inc. (Newburyport, MA). 1-[Bis(dimethylamino)methylene]-1H-1,2,3-triazolo[4,5-b] pyridinium 3-oxidhexafluorophosphate (HATU) was obtained from Chem Impex Intl Inc. (Wood Dale, IL), Optima MS-grade MeOH, HPLC-grade acetonitrile (MeOH), Optima MS-grade water, cell-culture-grade water (endotoxin-free), and 10 mg/mL gentamicin reagents were purchased from Fisher Scientific (Hampton, NH). Monoclonal mouse anti-human HIV-1 p24 (clone Kal-1, 1:200 dilution) and polymer-based horseradish-peroxidase-conjugated anti-mouse EnVision + secondary were purchased from Dako (Carpinteria, CA, IHC-Prediluted). Heat-inactivated pooled human serum was purchased from Innovative Biologics (Herndon, VA). Dulbecco's Modified-Eagle's Media (DMEM) was purchased from Corning Life Sciences (Tewksbury, MA). Flash chromatography was performed using 32–63 μm flash silica gels from SiliCycle Inc. (Quebec, Canada). Reactions were monitored by thin-layered chromatography on precoated silica plates (250 μm, F-254) from SiliCycle Inc. (Quebec, Canada). The compounds were visualized by UV fluorescence. CEM-ss CD4+ T cells were obtained from the HIV Reagent Program.

**Synthesis of L-phenylalanine and L-alanine docosyl esters**. N-(carbobenzy-loxy)- L- phenylalanine (2 g, 1 equiv.), or N-(carbobenzyloxy)- L- alanine (2 g, 1 equiv.), and 1-docosanol (1.1 equiv.) were suspended in anhydrous DMF (20 ml) and CHCl₃ (20 mL) and cooled to −10 °C on a regular ice/ NaCl bath. To this mixture, HATU (1.5 equiv), and imidazole (1 equiv) were then added. Next, Et₃N (2 equiv) was added to the pre-cooled mixture (−10 °C) dropwise. The resultant solution was stirred and gradually warmed to 45 °C over 24 h under an inert argon atmosphere. The reaction mixture was concentrated on a rotary evaporator to yield a light-yellow solid that was purified by silica column flash chromatography eluting with a 4:1 mixture of hexanes and ethyl acetate. Next, the Cbz-protected amino acid esters were dissolved in 30 ml CHCl₃ and 30 ml MeOH, and deprotected using palladium on carbon (Pd/C, 40% weight of N-Cbz- amino acid) and Et₃Si (15 equiv). After 24 h, the reaction was filtered through celite, concentrated, and precipitated from diethyl ether to yield amorphous white powders of the desired free amino acid ester compounds that were used directly in the subsequent steps.

**Synthesis and characterization of TFV ProTides**. Monophenyl tenofovir (2 g, 5.5 mmol, 1 equiv) was dried from anhydrous benzene (15 mL), resuspended in anhydrous ACN (25 mL), and then cooled at −10 °C under an argon atmosphere. After the addition of 5 equivalents of SOCl₂ the reaction was heated at 65 °C for 2 h under protection from light. The mixture was then dried from 15 ml benzene by rotary evaporation. The chlorinated compound was then resuspended in 15 ml of ACN and 15 ml THF, and cooled to −10 °C. A solution of L-phenylalanine docosyl ester (2 g, 4.22 mmol, 1 equiv), or L-alanine docosyl ester (2 g, 5.06 mmol, 1 equiv) was then added followed by dropwise addition of Et₃N (4 equiv) at −10 °C. The reaction mixture was then warmed to room temperature, then heated at 45 °C for 24 h under protection from light. The mixture was then concentrated to remove solvents. The residue was purified by silica column flash chromatography, eluting with 95% then 92.5% DCM in methanol. The desired compound fractions from the columns were dried on a rotary evaporator and freeze-dried from a mixture of diethyl ether/ water to obtain white powders, with chemical yields of 50–65%. FTIR analysis was performed on a Spectrum Two FT-IR spectrometer (PerkinElmer, Waltham, MA, USA).

**MALDI-TOF analysis**. Samples were re-suspended in TFA (30% Acetonitrile and 0.1% Trifluoroacetic acid) and diluted to ≤1 ug/ul. The MALDI matrix 2,5-Dihydroxybenzoic Acid (DHB) was re-suspended in TA30 with a working concentration of 40ug/ul and combined with the sample in a 1:1 ratio that was spotted on a Bruker MTP 384 ground steel target plate and allowed to dry. All spectra were recorded using a Bruker Autoflex Max MALDI-TOF mass spectrometer equipped with a smartbeam-II solid-state laser operating in positive reflectron mode and controlled with Bruker's flexControl (V3.4) instrument software. For each sample, spectra were acquired within a mass range of 500−3,500 Da at an acquisition rate of 2,000 Hz. The remaining instrument parameters were configured as follows: ion source 1 = 19.00 kV, ion source 2 = 16.75 kV, lens = 8.50 kV, reflector = 21.00 kV, reflector 2 = 9.60 kV, pulsed ion extraction time = 130 ns, and matrix suppression ≤ 450 Da. All raw spectra were processed with Bruker's flexAnalysis (V3.4) to include automated baseline subtraction and smoothing.

**Ultraperformance liquid chromatography–ultraviolet/visible drug and pro-drug quantifications**. A Waters ACQUITY ultraperformance liquid chromato-graphy (UPLC) H-Class system with a UV detector and Empower 3 software was used to measure the drug concentrations. M1TFV and M2TFV samples were separated on a Phenomenex Kinetex 5 µm C18 column (150 × 4.6 mm). TAF samples were separated on a Phenomenex Kinetex 5 µm C18 column (150 × 4.6 mm). TAF was detected at 260 nm using an isocratic elution with a mobile phase of 40% 7.5 mM ammonium acetate pH 4/60% MeOH with a flow rate of 0.25 ml/min. M1TFV and M2TFV were detected at 210 and 254 nm using an isocratic elution with a mobile phase of 98% MeOH/2% 7.5 mM ammonium bicarbonate pH 7.0 at a flow rate of 0.2 ml min. The drug content was determined relative to the peak areas of drug standards (0.05–50 µg/ml) in MeOH.

**Photodiode-array detection (PDA) assessment of purity**. A Waters Acquity H-Class Plus UPLC with PDA detector was used to qualitatively assess compound purity. M1TFV/M2TFV Method: 98% HPLC-grade Methanol, 2% 7.5 mM Ammonium Bicarbonate pH 7.0; flow 1.0 mL/min; column − Phenomenex Kinetex 5 µm C18 100 A, 150 × 4.6 mm with Phenomenex SecurityGuard and C18 4 × 3.0 mm guard cartridge; 3D data with lambda 190−490 nm, resolution 1.2 nm, sampling rate 10 points/sec on a Waters Acquity H-Class Plus UPLC with PDA detector. TAF Method: 60% HPLC-grade Methanol, 40% 7.5 mM Ammonium Acetate pH 4.0; flow 0.500 mL/min; column − Phenomenex Kinetex 5 µm C18 100A, 150 × 4.6 mm with Phenomenex SecurityGuard and C18 4 × 3.0 mm guard cartridge; 3D data with lambda 190−490 nm, resolution 1.2 nm, sampling rate 10 points/s on a Waters Acquity H-Class Plus UPLC with PDA detector.

**Solubility**. The aqueous and 1-octanol solubility of TAF, M1TFV, and M2TFV were determined by adding an excess of drug to water or 1-octanol in amber glass vials at room temperature and placed on a Thermo Scientific 4625 Plate Shaker at 100 rpm for 48 h. Samples were centrifuged at 20,000 × g for 10 min to separate the

solubilized drug from insoluble pellet. Aqueous supernatants were frozen, lyo-philized, and then resuspended in MeOH. 1-octanol supernatants were prepared for analysis by dilution in MeOH and samples analyzed for drug content by UPLC-UV (Waters Acquity H Class).

**Nanoparticle synthesis and characterization**. NM1TFV, NM2TFV, and NTAF solid drug nanoparticles were manufactured by high-pressure homogenization in aqueous buffers stabilized by non-ionic surfactants. The prodrug to surfactant ratio was maintained at 2:1 (w/w), and the suspension concentration was in the range 0.1–10% w/v for the drug/prodrug and 0.05–5% w/v for P407. Both NM1TFV and NM2TFV nanoformulations were prepared in P407 surfactant solution in 10 mM HEPES at pH 7. For NTAF, a mixture of P407 and PEG 3350 was used as stabi-lizers. Attempts to formulate TAF using P407 without PEG 3350 produced unstable formulations. Specifically, TAF was dispersed in a P407/PEG 3350 solu-tion in 10 mM HEPES pH 5.5 and allowed to form a pre-suspension. The prodrug to P407 to PEG 3350 ratio was maintained at 4:1:1 (w/w), and the suspension concentration was in the range 0.1–10% w/v for the drug/prodrug and 0.025–2.5% w/v for P407/PEG. The pre-suspensions were homogenized on an Avestin EmulsiFlex-C3 high-pressure homogenizer at 15,000−20,000 psi until the desired particle size of 250–350 nm was achieved. The homogenized solid drug nano-particles were then evaluated for particle size, homogeneity, and surface charge by dynamic light scattering using a Malvern Zetasizer Nano-ZS. Long-term nano-particle stability for all the formulations during storage was evaluated at room temperature over a period of 3 months. The amount of TAF, M1TFV, and M2TFV within the formulations after homogenization was determined from diluted for-mulation samples in MeOH (1,000–10,000-fold dilution) and analyzed by UPLC–ultraviolet/visible (UV/vis) spectroscopy using calibration curves with known standards. The percentage of drug entrapped within each nanosuspension was calculated using the equation: encapsulation efficiency (%) = (weight of the drug in formulation/initial weight of drug added) × 100.

**Prodrug chemical stability**. The pH buffers included 0.1% formic acid (pH 2.0), 7.5 mM ammonium acetate buffer (pH 6.0 adjusted with acetic acid), PBS (pH 7.0), 7.5 mM ammonium bicarbonate buffer (pH 8.0 adjusted with acetic acid), and 0.1% ammonium hydroxide (pH 10.3 and stored at 4 °C until use. NM1TFV or NTAF were incubated at 37 °C in 100 µL of the buffer. The final concentration of the drug in the incubation mixture was 1 µM. The reactions were stopped by the addition of 900 µL of acidified methanol (0.1% formic acid + 2.5 mM ammonium formate) at 0, 5 s, 30 min, 1, 2, 6, 12, and 24 h time points. Control samples were incubated using the same method as in the absence of substrates, however, substrates were added after the addition of acidified methanol. The mixtures were centrifuged at 16,000 × g for 10 min. The supernatants were aspirated and stored at −80 °C until ultra-performance liquid chromatography tandem mass spectrometry (UPLC-MS/MS) analysis.

**Human monocyte-derived macrophage (MDM) isolation, cultivation, and cytotoxicity**. Human monocytes were obtained, cultured, and differentiated into MDM as previously described[11,12,34,62]. The cell viability after treatment with nanoparticles was evaluated using the MTT assay. Human MDMs plated in 96-well plates at a density of 0.08 × 10⁶ cells/well were exposed to a range of concentrations (1.5625−200 µM) of NM1TFV, NM2TFV, or TAF for 8 h. The cells were then washed with PBS after which 100 µL of MTT solution was added to each well (5 mg/mL) for 45 min at 37 °C. The MTT solution was then replaced with 200 µL/ well of DMSO. A Molecular Devices SpectraMax M3 plate reader with SoftMax Pro 6.2 software (Sunnyvale, CA) was used to record absorbance at a wavelength of 490 nm.

**Drug nanoformulation uptake and retention**. Human MDMs were used for in vitro formulation assessments. MDM formulation uptake and retention studies were performed in flat-bottom, 12-well plates at a density of 1.2 × 10⁶ cells/well, with each treatment completed in triplicate. For cellular uptake studies, MDMs were treated with 10 µM or 100 µM NM1TFV, NM2TFV, and TAF. At 2, 4, and 8 h after treatment, the cells were collected, processed, and analyzed for drug content by UPLC-UV/vis as previously described[12,13,56,63]. For the retention studies, MDMs were treated with 10 µM or 100 µM NM1TFV, NM2TFV, or NTAF for 8 h, and then washed twice with 1 ml PBS. Then, a culture medium (without drug) was added, and a half-media volume was replaced every other day. MDMs were col-lected at days 1, 5, 10, 15, and 30 after treatment, and then analyzed for intra-cellular drug content by UPC-UV/vis. For TEM imaging of intracellular nanoparticles, MDMs were treated with 10 µM NM1TFV, NM2TFV, or TAF for 8 h and then washed twice with PBS. MDMs were then collected immediately after the 8 h treatment and analyzed by an FEI Tecnai G2 Spirit transmission electron microscope as previously described[11,12,14].

**Antiretroviral efficacy**. To determine long-term antiretroviral efficacy following a single treatment, MDMs were plated in flat-bottom 12-well plates at a density of 1.2 × 10⁶ cells/well. MDMs were treated with 10 µM or 100 µM NM1TFV, NM2TFV, NTAF, or TAF for 8 h. After treatment, the cells were washed twice with PBS and cultured in a fresh culture medium without drug with half-media

replacement every other day. On days 1, 5, 10, 15, 20, 25, and 30 following treatment, the cells were infected with HIV-1$_{ADA}$ (a macrophage-tropic viral strain) at a multiplicity of infection of 0.1 infectious particles/cell for 8 h. MDM were cultured, and media samples collected on day 10 for the measurement of HIV reverse transcriptase (RT) activity as previously described[64,65]. Cells were fixed in 2% PFA at each time point, and expression of the HIV-1p24 antigen was determined by immunocytochemistry. To determine the IC$_{50}$ of each prodrug and its nanoformulation, MDMs were plated in flat-bottom 96-well plates (0.08 × 10$^6$ cells/well). Cells were treated with a range of drug concentrations, 0.00005–5,000 nM, of TAF, M1TFV, M2TFV, NTAF, NM1TFV, or NM2TFV for 1 h prior to infection with HIV-1$_{ADA}$ (multiplicity of infection of 0.1) for 4 h. After 4 h of viral challenge, the cells were washed with PBS and given fresh media that contained the same concentrations of drug (0.00005–5,000 nM). Cell supernatants were collected 10 days later and assayed for HIV-1 RT activity as described above. Replicate studies were performed in CEM-ss CD4+ T-cell cultures. Specifically, cells were plated in 96-well plates, centrifuged at 650 × g for 5 min, and resuspended in (0.1–1,000 nM) drug-containing media for 1 h. The subsequent challenge with HIV-1$_{NL4–3–eGFP}$, cell culture, sample collection, and determination of HIV-1 RT activity were performed as previously reported[13,15].

**PK studies.** Twelve-week-old healthy Sprague Dawley rats (male, 300–350 g; female, 200–230 g; SASCO) were purchased from Charles River Laboratories (Wilmington, MA) and housed in the University of Nebraska Medical Center (UNMC) laboratory animal facility according to the American Animal Association and Laboratory Animal Care guidance. All procedures were approved by the Institutional Animal Care and Use Committee at the University of Nebraska Medical Center (UNMC) as set forth by the National Institutes of Health (NIH). Following a one-week quarantine, the animals were randomized by weight and administered a single intramuscular injection (IM, caudal thigh muscle; 200 μL) of 75 mg/kg TFV equivalents of either NM1TFV, NM2TFV, or NTAF (males) or NM2TFV or NTAF (females). On days 1, 14, 28, 42, and 56 after injection, blood samples were collected from non-fasted animals into EDTA tubes via the retro-orbital sinus using autoclaved glass capillaries. A 25 μL aliquot of blood was immediately diluted into 1 mL MeOH (containing 2.5 mM ammonium formate and 0.1% formic acid) for drug quantitation. The remaining 500 μL−1mL of whole blood was used for PBMC isolation. On day 28 (females) or days 7, 28, and 56 (males) post-drug administration, non-fasted animals were humanely euthanized. Beginning in the morning, all animals treated with NM1TFV were first terminated, followed by all NM2TFV, then all NTAF. Spleen, lymph node, adipose, testes, liver, lung, gut (duodenum/jejunum), kidney, brain, vaginal and rectal tissue were collected for the quantitation of TFV and prodrugs. Additionally, rat PBMCs, splenocytes, lymphocytes, and liver parenchymal and non-parenchymal cells were isolated on sacrifice days for analysis of TFV-DP content. The amount of TFV, M1TFV, and M2TFV in blood and tissues were quantified from calibration curves with known standards by UPLC–MS/MS using a Waters ACQUITY H-class UPLC connected to a Xevo TQ-S micro mass spectrometer. The solvents used for sample processing and analytical methods were MS-grade (Fisher). Serum chemistry profile, animal body weights, organ to body weight ratios, complete blood counts (CBCs), and histological evaluations were used to assess adverse formulation reactions. Histological examination was performed on 5 μm tissue sections stained with hematoxylin and eosin staining. The images were captured on a Nuance EX multispectral imaging system affixed to a Nikon Eclipse E800 microscope (Nikon Instruments) and evaluated by a certified pathologist in accordance with the guidelines of the Society of Toxicologic Pathology. Serum chemistry data sets were acquired on a VetScan comprehensive diagnostic profile disc and a VetScan VS-2 instrument (Abaxis Veterinary Diagnostics). Biochemical analyses of CBCs were performed using an Abaxis Vetscan HM5 Hematology Analyzer. For both, the results for treated animals were compared with those from age- and sex-matched untreated control rats.

**TFV-DP sample preparation.** Intracellular TFV-DP was extracted from human MDM or rat PBMCs as follows. MDM were treated with 10 μM or 100 μM NM1TFV, NM2TFV, or TAF. At 2, 4, and 8 h after treatment, MDM was washed with PBS to remove the excess free drug. The cells were then collected in 70% MS-grade MeOH. TFV-DP from MDM was extracted as follows. Sep-Pak QMA cartridges (360 mg, 37–55 μm; Waters, Milford, MA) were used to separate TFV-DP from TFV- and mono-phosphate counterparts. The QMA cartridges were conditioned with 10 ml of 500 mM KCl, followed by 10 ml of 5 mM KCl. The sample (200 μL) was loaded onto the cartridges and washed with 12 ml of 75 mM KCl. The triphosphate (TP) fraction was eluted with 3 ml of 500 mM KCl. The pH of the TP fraction was lowered to 4.25 by adding 45 μl ammonium acetate buffer (pH 4, 10 mM) per ml eluate, and dephosphorylated by adding one unit of type XA sweet potato acid phosphatase (Sigma-Aldrich) per ml eluate (3 μL per sample) and incubating at 37 °C for 45 min. 150 μL of 12% TFA and 10 μL of 5 ng/mL $^{15}$N$_2$ $^{13}$C-3TC and d4- ABC internal standard were then added. Samples were then loaded onto Waters OASIS HLB cartridges (60 mg, 30 μm; Waters, Milford, MA) pre-conditioned with 3 ml 100% methanol followed by 3 ml 1% TFA to remove salts. Salts were removed from the dephosphorylated samples with 3.5 ml of 1% TFA in water, then eluted with 1.5 ml 100% methanol and evaporated under vacuum. Once dry, the residue was stored at −80 °C until UPLC–MS/MS analyses.

Samples were reconstituted in 100 μL of 10% MeOH, 90% 7.5 mM ammonium bicarbonate pH 7 in LC-MS grade water.

**PBMC isolation from whole blood.** PBMCs were isolated from ~500 μL to 1mL volumes of whole blood collected by retro-orbital sinus punctures then placed into EDTA-coated tubes. The whole blood was layered on top of an equal volume of Histoplaque®-1083 (Sigma-Aldrich) at room temperature and centrifuged at 400 × g for 30 min at room temperature with an off-brake setting. The PBMCs were then collected from the plasma/Histoplaque interface into a 5 mL conical centrifuge tube containing 3 mL sterile PBS and centrifuged at 250 × g for 10 min at room temperature. Following aspiration of the supernatant, the pellet was resuspended in 1.5 mL of sterile PBS and centrifuged at 250 × g for 10 min. The procedure was repeated twice. Following cell counting, the PBMCs were lysed in 200 μL of LC-MS grade 70% MeOH, 30% water, and stored at −80 °C.

**TFV-DP extraction from mucosal tissues.** On day 28 (female rats) or days 28 and 56 (male rats) post drug treatment, rats were humanely euthanized. Rectal and vaginal tissues were collected, flash-frozen in liquid nitrogen, and stored at −80 °C until further processing. For TFV-DP quantitation, vaginal and rectal tissues were homogenized in 50% and 70% MeOH in water, respectively, using a Qiagen TissueLyser II. TFV-DP was extracted from the homogenates. Individual cell populations were not isolated from the tissues. TFV-DP samples were further processed and analyzed as described herein.

**Sample preparation and UPLC-MS/MS analyses.** TFV, M1TFV, and M2TFV in biological samples were quantitated by UPLC-MS/MS using a Waters Acquity H-class UPLC connected to a Waters Xevo TQ-S micro mass spectrometer (Milford, MA). For blood and tissue analysis of TFV, 10 μL of internal standard (IS) solution (50 ng/ml 3TC-d3) was added to each sample. Lymph node and spleen samples were homogenized in nine volumes of 90% MeOH (0.1% Formic acid + 2.5 mM ammonium formate), while all other tissues were homogenized in four volumes of 90% MeOH (0.1% Formic acid + 2.5 mM ammonium formate). 1 mL MeOH (0.1% Formic acid + 2.5 mM ammonium formate) was added to 25 μL of whole blood to extract TFV. Samples were separated on a CSH C18 (1.7 μm, 2.1 × 100 mm) column using a mobile phase consisting of 7.5 mM ammonium bicarbonate adjusted to pH 7 (mobile phase A) and 100% Optima MS-grade methanol (mobile phase B) at a flow rate of 0.22 mL/min and run time of 8 min. TFV was detected in the positive ionization mode using multiple reaction monitoring (MRM) transitions for TFV and 3TC–d3, of 288.23 > 159.01, and 233.23 > 114.97 m/z, respectively. The initial mobile phase composition was 5% B for the first 4.25 min at which time it was increased to 30% B over 0.25 min, then increased to 90 % B and held for a further 0.6 min before returning to initial conditions. Spectra were analyzed and quantified by MassLynx software version 4.1. Quantitation was based upon drug peak area to internal standard peak area ratios.

For quantitation of M1TFV and M2TFV in blood and tissues, 10 μL of internal standard (IS) solution (500 ng/ml of stearoylated darunavir (SDRV)) was added to each sample. Lymph node and spleen samples were homogenized in nine volumes of 90% MeOH (0.1% Formic acid + 2.5 mM ammonium formate), while all other tissues were homogenized in four volumes of 90% MeOH (0.1% Formic acid + 2.5 mM ammonium formate). 1 mL MeOH (0.1% Formic acid + 2.5 mM ammonium formate) was added to 25 μL of whole blood to extract prodrug. Chromatographic separation of 10 μL sample injections of M1TFV/ M2TFV was achieved with an ACQUITY UPLC-BEH Shield RP18 column (1.7 μm, 2.1 × 30 mm) using a gradient of mobile phase A (7.5 mM ammonium formate in Optima-grade water adjusted to pH 3 using formic acid) and mobile phase B (100% Optima-grade methanol) at a flow rate of 0.35 mL/min. The initial mobile phase composition was 88% B for the first 5 min at which time it was increased to 95% B over 0.25 mins and held constant for 1.50 min. Prodrugs were detected in the positive ionization mode using MRM transitions for M1TFV, M2TFV, and SDRV, of 819.70 > 206.10, 743.68 > 206.10, and 814.70 > 155.96 m/z, respectively. Spectra were analyzed and quantified by MassLynx software version 4.1. Quantitation was based upon drug peak area to internal standard peak area ratios.

TFV-DP: Parent TFV was generated from TFV-DP during sample preparation and then subjected to UPLC-MS/MS analyses. The UPLC-MS/MS system was comprised of a Waters ACQUITY UPLC system coupled to a Waters Xevo TQ-XS triple quadrupole mass spectrometer (Waters, Milford, MA) with electrospray ionization (ESI) source. A CSH C18 analytical column (2.1 × 100 mm, 1.7 μm) equipped with a guard column (Waters, Milford, MA) at 30 °C was used for analyte separation. Samples were maintained at 4 °C during analysis. The MRM transitions of 288.10 < 159.04 m/z and 232.97 < 115.00 m/z were used for TFV and 3TC-d3 internal standard (IS) quantification, respectively. Mobile phase A consisted of 7.5 mM ammonium bicarbonate in water (MS grade, Fisher) with pH adjusted to 7.0 with glacial acetic acid (ACS grade, Sigma). Mobile phase B was 100% methanol (MS grade, Fisher). The flow rate was 0.22 mL/min. A gradient of 95% A and 5% B was held for 4.25 min, and then B was increased to 30% over 0.25 min. B was then held at 90% for 0.4 min before returning to initial conditions. Spectra were analyzed and quantified by MassLynx software version 4.1. Quantitation was based upon drug peak area to internal standard peak area ratios.

**Statistical analyses**. Statistical analyses of data sets were performed on a GraphPad Prism v9.0.0.0 software and Microsoft Excel. Both in vitro and in vivo study data sets are expressed as means ± SEM with a minimum of three biological replicates for each study. The Student's t-test (two-tailed, paired, and unpaired) was used to compare means between two groups while a one-way ANOVA followed by Tukey's post hoc test was used for comparisons between three or more groups. Statistical significances were denoted as *$P < 0.05$, **$P < 0.01$, ***$P < 0.001$, and ****$P < 0.0001$. Comparison between intramuscular and tissue drug concentrations was performed by Pearson correlation and linear regression models.

**Study approvals**. All animal studies were approved by the University of Nebraska Medical Center Institutional Animal Care and Use Committee in accordance with the standards incorporated in the *Guide for the Care and Use of Laboratory Animals*[66]. Human monocytes were isolated by leukapheresis from HIV-1/2 and hepatitis B seronegative donors according to an approved University of Nebraska Medical Center Institutional Review Board exempt protocol. All donors gave informed consent for the use of the deidentified material.

**Reporting summary**. Further information on research design is available in the Nature Research Reporting Summary linked to this article.

## Data availability

The data supporting the study's findings are available within the article and its supplementary files. All the relevant data used to generate main and supplementary figures are included as Source Data (https://doi.org/10.6084/m9.figshare.15168909). Source data are provided with this paper.

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

## Acknowledgements

We thank the University of Nebraska Medical Center (UNMC) cores for NMR (Ed Ezell), Elutriation and Cell Separation (Myhanh Che, Na Ly, and Li Wu), Electron Microscopy (Nicholas Conoan and Tom Bargar), Histology (Tissue Sciences Facility), MALDI/TOF Mass Spectrometry (Eric Troudt), compound purity by UPLC photodiode array detector (Brady Sillman), as well as Comparative Medicine for technical assistance and animal care. We also wish to thank Gilead Sciences Inc. for the supply of TAF. This research is supported by the University of Nebraska Foundation, which includes donations from the Carol Swarts, M.D. Emerging Neuroscience Research Laboratory, the Margaret R. Larson Professorship, the Frances and Louie Blumkin Endowment, and the Harriet Singer Endowment; the Vice Chancellor's Office of the University of Nebraska Medical Center for Core Facilities; Nickolus Badami Fellowship (to B.E.) and the National Institutes of Health grants 1R01AI145542-01A1, 1R56 AI138613-01A1, P01 DA028555, and P30 MH062261.

## Author contributions

D.C.—study design, synthesized the prodrugs and formulations, design and execution of most experiments, data acquisition, data analysis, and interpretation, co-wrote paper; N.S.—data acquisition and interpretation; S.D.—data acquisition and interpretation; A.B.—assisted with animal study design and data acquisition; B.L.S.D.—performed LC-MS/MS analyses; J.M.—designed, supervised, and analyzed the mass spectrometry data; N.G.—development of LC-MS/MS methods, data acquisition and interpretation; Y.A.—development of LC-MS/MS methods, data acquisition, and interpretation; S.M.C.—acquisition and interpretation of toxicology data sets, histopathology, and TEM examinations; H.E.G.—design of experiments, data interpretation, supervision of experiments, co-wrote the paper, and funding acquisition; B.E.—conceived project, study design, design of synthesis and formulation experiments, supervision of experiments, data analysis and interpretation, co-wrote the paper and funding acquisition. All authors critically evaluated the paper prior to submission.

## Competing interests

B.E. and H.E.G are cofounders of Exavir Therapeutics, Inc. and are inventors on patents that cover stable ProTides and the long-acting slow effective release formulations. The authors declare that this work was produced solely by the authors and that no other individuals or entities influenced any aspects of the work including, but not limited to, the study conception and design, data acquisition, analyses, and interpretation, and writing of the paper. The authors further declare that they have received no financial compensation from any other third parties for any aspects of the published work. The remaining authors declare no competing interests.
