## [Peer Review File · Nature Communications]

REVIEWER COMMENTS

Reviewer #1 (Remarks to the Author):

This is an innovative and comprehensive manuscript that develops tenofovir (TFV) into long acting hydrophobic and lipophilic ProTides to form aqueous poloxamer stabilized prodrug nanocrystals. Very thorough study

- 1) Cell culture experiments. What was 8 hr used for treatment for MTT assay. Was long term cell viability examined.
- 2) Please explain the rationale for 8 hr uptake studies as well
- 3) Please clarify P407/Peg 3350 ratios and why it is necessary or TAF
- 4) it would be interesting to see what happens with separate injection at 56 days. Will mast cells occur or was a IgG or IGM response measured.

Reviewer #2 (Remarks to the Author):

The article describes the development of two lipophilic ProTides to form aqueous poloxamer stabilized prodrug nanocrystals as long-acting drugs for antiviral therapy. the prodrugs synthesis, their nanocrystal formulations, and their stability at different pH, and in vitro and in vivo experiments to assess the antiviral activity toxicity are reported.

The author(s) found that one of the prodrug NM1TFV significantly improved drug and TFV-DP biodistribution compared to NTAF

The results are interesting, and the topic is certainly in the remit of the Journal Nature Communications.

However, the author(s) are suggested to address the following comments as mentioned:

- 1) The ProTide activation starts with a first enzymatic activation leading to the cleavage of the ester mediated by cathepsin Y enzyme. Is this happening also with this lipid ProTides version? Any evidence of that?
- 2) Page 7 line 44 states: "However, TAF is susceptible to hydrolytic degradation." this should be referenced and few more details about the hydrolytic instability of TAF should be added. In the body? Where in the body?
- 3) It would be good to have the high-resolution mass for the final compounds. Although indicated as HRMS (ESI-TOF) the value reported is a Low resolution mass.
- 4) From the phosphorus nmr of M1TDF in the SI and its proton nmr in the manuscript Fig E another compound TDF related seems to be present (small peak in the ^{31}P between the 2 diastereoisomers and the small peaks at around 8 in ^1H). The purity of the final compounds and the method used to assess it should be reported and should be $>95\%$
- 5) page 13 line 314-315 states: "The triplets at 7.10, 7.15 and 7.31 ppm represent the aryl and phenol groups (Fig. 1e). The signal at 7.10, 7.15 and 7.31 cannot be triplet most probably are due to doublet overlapping between the 2 diastereoisomers. These need to be corrected at page 13 and in the SI where the description of the nmr is reported.
- 6) page 13 line 314-315 states: "Peak splitting in the phosphorous NMR spectra stereoisomers indicated a 1:1 mixture of the R and S (Supplemental Fig. 1b)" this is not a splitting they are two separated signals for the 2 diastereoisomers. the sentence should be rephrased
- 7) The captions for nmr spectra need to include the deuterated solvent and the NMR instrument used.
- 8) For determining the aqueous solubility what does it mean "mixing for 48 h" at page 7 line 158. Mixing how? the exact method should be described.

Reviewer #3 (Remarks to the Author):

In the manuscript "Transformation of tenofovir into stable ProTide nanocrystals with long-acting pharmacokinetic profiles" authors described a development of a long-acting formulation of TFV. They created new TFV-based prodrugs with phenylalanyl and alanyl docosyl esters that were

manufactured to nanoparticles by high pressure homogenization and stabilized by non-ionic surfactants. These formulations showed uptake into primary monocyte-derived macrophages, inhibited HIV-RT activity, and prevent cell infection in vitro. PK analysis showed that a single intramuscular injection of 75 mg/kg TFV equivalent of new formulations to rats resulted in TFV-DP levels above 4xIC₉₀ in blood leucocytes for 2 months. New formulations have strong potential for development of long-acting tenofovir.

Although manuscript is well written, several inconsistencies need to be addressed:

1. Line 35: primary human CD4 T cells were not used for the uptake assay, instead authors used a human T4-lymphoblastoid cell line CEM-SS. Please clarify this.
2. Line 36: Specify that HIV challenges were done in vitro.
3. Lines 207-210: This part looks like repetition of previous text, please specify that this describes TEM imaging.
4. Lines 212-230: In the paragraph "Antiviral efficacy", several experiments are described, please specify. It seems that lines 212-220 describe in vitro HIV challenges, lines 220-230 IC₅₀ determination.
5. Lines 222 and 225: range of concentrations used for MDM is 0.00005-5000nM, but in the Figure 2c legend, the range is 0.0001-1000nM. Please explain/unify.
6. Line 238: How were animal randomized, i.e. weight, sex etc.?
7. Line 242: Specify how much blood was collected, alternatively, how much blood was used for PBMC isolation (line 244).
8. Line 244: Provide brief description how were PBMCs isolated and blood volume used.
9. Lines 282-283: Based on these lines it seems that alanine aa ester was used for M1TFV and phenylalanine aa ester for M2TFV. However, in the line 286, opposite order is provided. Clarify what M1 TFV and M2TFV are. For clarity, specify which structure is M1 TFV and M2TFV in the Figure 1b.

We are pleased to respond to each of the queries raised in review point-by-point. The required amendments made in the text are highlighted in yellow.

Reviewer 1

“This is a innovative and comprehensive manuscript that develops tenofovir (TFV) into long acting hydrophobic and lipophilic ProTides to form aqueous poloxamer stabilized prodrug nanocrystals. Very thorough study.”

Response: We agree. Thank you very much.

Point 1. *“Cell culture experiments. What was 8 hr used for treatment for MTT assay. Was long term cell viability examined.”*

Response: An 8h incubation period reflects the drug nanoparticle uptake, retention, and antiviral efficacy that would occur during the early stages of particle dissemination from the site of injection to the reticuloendothelial system in monocyte-macrophages. For these studies, human monocyte-derive macrophages (MDM) are incubated with the test drug particle for 8h followed by a wash out. This reflects the earliest events of the drug nanoparticle incubation and the time most likely to affect cytotoxicity. To this end the in vitro studies were focused on 8h. In this assays cell vitality was measured through the mitochondrial MTT cell vitality assay by conversion of the tetrazolium dye 3-(4,5-dimethylthiazol-2-yl)-2,5-diphenyltetrazolium bromide to its insoluble formazan. The assay measures cellular (macrophage) metabolic activity and is used as an indicator of cell function, viability and cytotoxicity. Long-term cell viability is reflective of virus protection during serial challenge with HIV-1. This fact is now clarified in the methods section.

Point 2. *“Please explain the rationale for 8 hr uptake studies as well.”*

Response: Prior extensive work from our laboratory has demonstrated early cytotoxicity occurs as a result of macrophage activation, secretion of bioactive inflammatory factors and infiltration to the injection sites within hours of drug particle injections. The drug nanoparticle uptake in macrophages occurs within 8 h (*Nanotoxicology*. 2011 Dec;5(4):592-605; *Nanomedicine (Lond)*. 2009 Dec;4(8):903-17). This is now clarified and referenced, lines 172-173. An 8 h incubation period was also used for the cell retention and long-term antiretroviral efficacy studies, therefore 8 h incubation period was used in the uptake study for consistency of in vitro assays.

Point 3. *“Please clarify P407/Peg 3350 ratios and why it is necessary or TAF”*

Response: We used a 4:1:1 TAF to P407 to PEG 3350 ratio to produce a stable NTAF formulation. This ratio was selected because the 2:1 drug to P407 ratio that produced stable NM1- and NM2TFV nanocrystals without a PEG 3350 additive generated unstable TAF nanocrystals, presumably due to differences in compound physicochemical properties that affects how various nonionic formulation surfactants stabilize solid drug nanoparticles through surface coating and steric interactions. This is now clarified in the text, supplement lines 116-117.

Point 4. *“It would be interesting to see what happens with separate injection at 56 days. Will mast cells occur or was a IgG or IGM response measured?”*

Response: The transient presence of mast cells at the intramuscular site of injection is a normal physiological tissue injury response and has been previously observed after intramuscular injection of phosphate buffered saline or aqueous nanocrystalline compounds (*Journal of the Neurological Sciences* 135 (1996) 10- 17 and *Toxicologic Pathology* 2016, Vol. 44(2) 189-210). As hypothesized by Darville *et. al*, mast cells could affect host protein adsorption to the particle surface to influence phagocytosis efficiency, lymphatic nanoparticle uptake and prodrug hydrolytic susceptibility. Other findings by Paquette *et al*. noted that the foreign body response observed following intramuscular or subcutaneous dosing of aqueous micronized drug particles does not elicit an immune activation beyond the injection site (*Pharm Res.* 2014; 31(8): 2065–2077). Therefore, the early recruitment of mast cells may have initiated a cascade of events that ultimately underlie the observed extended drug half-life and sequestration in lymphoid and reticuloendothelial tissues. We did not measure IgG or IgM response; however, such immunogenicity markers were previously measured (*PLoS One.* 2015 Dec 30;10(12):e0145966. doi: 10.1371/journal.pone.0145966). In our prior studies we have assessed temporal leukocyte responses, drug uptake and distribution following multiple site injection of nanoformulated drugs. We found that local inflammatory responses heralded drug distribution to macrophage populations, regional lymph nodes, spleen and liver. This proceeded changes in myeloid populations as determined through flow cytometric tests of changed CD45, CD3, CD11b, F4/80, and GR-1 surface markers.

Reviewer 2

“The article describes the development of two lipophilic ProTides to form aqueous poloxamer stabilized prodrug nanocrystals as long-acting drugs for antiviral therapy. the prodrugs synthesis, their nanocrystal formulations, and their stability ad different pH, and in vitro and in vivo experiments to assess the antiviral activity toxicity are reported. The author(s) found that one of the prodrug NM1TFV significantly improved drug and TFV-DP biodistribution compared to NTAF The results are interesting, and the topic is certainly in the remit of the Journal Nature Communications. However, the author(s) are suggested to address the following comments as mentioned.”

Response: Thank you very much and as suggested the comments made below are fully addressed.

Point 1. *“The ProTide activation starts with a first enzymatic activation leading to the cleavage of the ester mediated by cathepsin Y enzyme. Is this happening also with this lipid ProTides version? Any evidence of that?”*

Response: We have not identified specific enzymes involved in our prodrug activation but do believe that M1 and M2 TFV ProTides would be susceptible to enzymatic hydrolysis. This hydrolysis would occur by cathepsins and carboxylesterases. Though, in the current report, we have not studied the biotransformation of M1- and M2TFV with purified enzymes there is evidence suggesting that enzymatic hydrolysis contributes to prodrug activation. Indeed, when M1TFV was incubated in human plasma, with or without heat-inactivation, greater prodrug stability is observed following heat-inactivation. This suggests that the lipophilic prodrugs are hydrolyzed, in large measure, by enzymes.

Point 2. *“Page 7 line 44 states: “However, TAF is susceptible to hydrolytic degradation.” this should be referenced and few more details about the hydrolytic instability of TAF should be added. In the body? Where in the body?”*

Response: The primary site for the hydrolysis of TAF is within lymphocytes, macrophages, and hepatocytes which contain carboxyesterases. However, degradation can also occur through non-enzymatic processes in aqueous buffers (*Antimicrob. Agents Chemother.* 2015, 59, 3913–3919 and *Journal of pharmaceutical and biomedical analysis* 131 (2016): 146-155). Additional details on TAF's hydrolytic instability supported by new references are incorporated into the revised manuscript, line 77.

Point 3. *“It would be good to have the high-resolution mass for the final compounds. Although indicated as HRMS (ESI-TOF) the value reported is a Low resolution mass.”*

Response: We agree and have now included high-resolution mass data sets into supplemental figures 1-3, lines 105-107.

Point 4. *“From the phosphorus nmr of M1TDF in the SI and its proton nmr in the manuscript Fig E another compound TDF related seems to be present (small peak in the 31p between the 2 diastereoisomers and the small peaks at around 8 in 1H. The purity of the final compounds and the method used to assess it should be reported and should be >95%”*

Response: We agree with the reviewer that the synthesized prodrugs are not 100% pure. We now have included compound purity data sets (see supplemental figures 5-7). The purity of the compounds was assessed by UPLC-UV/Vis. By comparing the total area of all peaks to the peak area of each compound, the average purity for M1TFV was 98.37%, 94.96% for M2TFV, and >99.9% for TAF hemifumarate (provided by Gilead Sciences).

Point 5. *“page 13 line 314-315 states: The triplets at 7.10, 7.15 and 7.31 ppm represent the aryl and phenol groups (Fig. 1e). The signal at 7.10, 7.15 and 7.31 cannot be triplet most probably are due to doublet overlapping between the 2 diastereoisomers. These need to be corrected at page 13 and in the SI where the description of the nmr is reported.”*

Response: We agree that the peak at 7.14 is a doublet. This is now corrected, lines 129-130. Also as shown below, the chemical shifts at 7.10 and 7.31 ppm represent triplets from the unsubstituted aryl and phenol groups.

Point 6. “page 13 line 314-315 states: Peak splitting in the phosphorous NMR spectra stereoisomers indicated a 1:1 mixture of the R and S (Supplemental Fig. 1b)” this is not a splitting they are two separated signal for the 2 diastereoisomers. the sentence should be rephrased”

Response: We agree with the reviewer that these are two separate peaks representing the two diastereomers at the phosphorous atom and the sentence has been rephrased to reflect this, line 131-132.

Point 7. “The captions for nmr spectra need to include the deuterated solvent and the NMR instrument used.”

Response: The NMR spectra were acquired in deuterated chloroform and these details are now included on each spectrum, figure 1.

Point 8. “For determining the aqueous solubility what does it means “mixing for 48 h” at page 7 line 158. Mixing how? the exact method should be described.”

Response: A Thermo Scientific 4625 Plate Shaker set to 100 rpm was used to mix the samples for 48 h. This has been added to the supplement, lines 103-104.

Reviewer 3

“In the manuscript “Transformation of tenofovir into stable ProTide nanocrystals with longacting pharmacokinetic profiles” authors described a development of a long-acting formulation of TFV. They created new TFV-based prodrugs with phenylalanyl and alanyl docosyl esters that were manufactured to nanoparticles by high pressure homogenization and stabilized by nonionic surfactants. These formulations showed uptake into primary monocyte-derived macrophages, inhibited HIV-RT activity, and prevent cell infection in vitro. PK analysis showed that a single intramuscular injection of 75 mg/kg TFV equivalent of new formulations to rats resulted in TFV-DP levels above 4xIC90 in blood leucocytes for 2

months. New formulations have strong potential for development of long-acting tenofovir. Although manuscript is well written, several inconsistencies need to be addressed”

Response: Thank you very much and the comments are now addressed.

Point 1. “Line 35: primary human CD4 T cells were not used for the uptake assay, instead authors used a human T4-lymphoblastoid cell line CEM-SS. Please clarify this.”

Response: The text has been updated to reflect the experimental conditions, line 35.

Point 2. “Line 36: Specify that HIV challenges were done in vitro.”

Response: Agreed. We now specify that the HIV challenges were done *in vitro*, line 36.

Point 3. “Lines 207-210: This part looks like repetition of previous text, please specify that this describes TEM imaging.”

Response: This section has been revised to clarify the TEM imaging procedures, supplement lines 164-165.

Point 4. “Lines 212-230: In the paragraph “Antiviral efficacy”, several experiments are described, please specify. It seems that lines 212-220 describe in vitro HIV challenges, lines 220-230 IC50 determination”

Response: Descriptors of both experiments have been added to the manuscript to distinguish the two studies: supplement lines 171,178-181.

Point 5. “Lines 222 and 225: range of concentrations used for MDM is 0.00005-5000nM, but in the Figure 2c legend, the range is 0.0001-1000nM. Please explain/unify”

Response: Thank you very much. 0.00005-5000nM is the correct concentration range for the test articles, the figure legend has been corrected.

Point 6. “Line 238: How were animal randomized, i.e. weight, sex etc.?”

Response: The animals were randomized by weight. Each treatment group consisted of animals of the same sex and age. The text has been updated with the method of randomization, supplement line 197.

Point 7. “Line 242: Specify how much blood was collected, alternatively, how much blood was used for PBMC isolation (line 244).”

Response: The amount of blood collected from each animal ranged from 500 μ L to 1mL. 25 μ L of whole blood was reserved for drug quantitation, and the remaining volume used to isolate PBMCs. The text has been updated with these details, supplement line 203.

Point 8. *“Line 244: Provide brief description how were PBMCs isolated and blood volume used.”*

Response: A description of the PBMC isolation was added to the supplemental materials. Specifically, PBMCs were isolated from ~500 µL-1mL volumes of whole blood collected by retro-orbital sinus punctures then placed into EDTA-coated tubes. The whole blood was layered on top of an equal volume of Histoplaque®-1083 (Sigma-Aldrich) at room temperature and centrifuged at 400 x G for 30 min at room temperature with off-brake setting. The PBMCs were then collected from the plasma/Histoplaque interface into a 5mL conical centrifuge tube containing 3 mL sterile PBS and the centrifuged at 250 x G for 10 min at room temperature. Following aspiration of the supernatant, the pellet was resuspended in 1.5 mL of sterile PBS and centrifuged at 250 x G for 10 min. The procedure was repeated twice. Following cell counting, the PBMCs were lysed in 200 µL of LC-MS grade 70% MeOH, 30% water and stored at -80°C.

Point 9. *“Lines 282-283: Based on these lines it seems that alanine aa ester was used for M1TFV and phenylalanine aa ester for M2TFV. However, in the line 286, opposite order is provided. Clarify what M1 TFV and M2TFV are. For clarity, specify which structure is M1 TFV and M2TFV in the Figure 1b.”*

Response: Thank you for noting this. Names have been added to each structure in Figure 1b for clarity. M1TFV is phenylalanine ester modified ProTide, while M2TFV has alanine ester. The text has been updated to be consistent with this, line 96.

We appreciate the opportunity to amend our manuscript based on the thoughtful reviews provided. We trust that it is now acceptable for publication in Nature Communications. Please do not hesitate to call on me again if further clarifications are needed.

Sincerely yours,

Benson Edagwa, PhD
Associate Professor
Department of Pharmacology & Experimental Neuroscience

REVIEWERS' COMMENTS

Reviewer #1 (Remarks to the Author):

Authors have addressed my concerns

Reviewer #2 (Remarks to the Author):

I am now happy with the revised manuscript and I recommend it for publication in Nature Communications.

Reviewer #3 (Remarks to the Author):

Authors answered all my comments and concerns